# Ameliorating Seed Germination and Seedling Growth of Nano-Primed Wheat and Flax Seeds Using Seven Biogenic Metal-Based Nanoparticles

**Maryam Bayat** [1], **Meisam Zargar** [1,*], **Kheda Magomed-Salihovna Murtazova** [2], **Magomed Ramzanovich Nakhaev** [3] and **Sergey I. Shkurkin** [4]

1   Department of Agrobiotechnology, Institute of Agriculture, RUDN University, 117198 Moscow, Russia; maryambayat1313@yahoo.com
2   Engineering Center of Carbon, Chechen State University, 364024 Grozny, Russia; fu.ggni@mail.ru
3   Applied Mathematics and Computer Technology, Chechen State University, 364024 Grozny, Russia; mr-nakhaev@mail.ru
4   All-Russian Scientific and Research Institute of Agrochemistry, Federal State Budgetary Institution, 344006 Moscow, Russia; s.shkurkin@sibac.ru
*   Correspondence: zargar_m@pfur.ru

**Abstract:** Recently, large-scale agriculture has led to increasing crop production. To increase crop productivity in large-scale cropping systems, attempts have been made to make nano-fertilizers and deliver them to the crops by extension of nanotechnology. Hence, nano-fertilizers might be defined as nanoparticles that may directly assist in supplying essential nutrients for crop productivity. Seed germination is the first and the most susceptible stage in the plant's growing phases, so could be considered as an index to evaluate the effect of newly developed materials such as nanoparticles (NPs), providing useful information for researchers. In our experiments, germination tests have been carried out in Petri dishes containing wet filter paper and nano-primed seeds. We had biosynthesized seven nanoparticles in our previous studies including calcinated and non-calcinated zinc oxide, zinc, magnesium oxide, silver, copper, and iron nanoparticles. The effect of these biogenic nanoparticles and their counterpart metallic salts including zinc acetate, magnesium sulfate, silver nitrate, copper sulfate, and iron (III) chloride was studied on two popularly grown plants, wheat and flax, in laboratory conditions to obtain preliminary information for future field experiments. Germination percentage, shoot length, root length, seedlings length, root–shoot ratio, seedling vigor index (SVI), shoot length stress tolerance index (SLSI), and root length stress tolerance index (RLSI) were calculated on the second and seventh days of the experiment. According to the results, the response of the plants to metal containing nanoparticles and metal salts mainly depend on the type of the metal, plant species, concentration of the NP suspension or salt solution, condition of the exposure, and the stage of growth.

**Keywords:** metal-based nanoparticles; biosynthesis; wheat; flax; seed germination; seedling growth; phytotoxicity





## 1. Introduction

The rapidly growing population and crop consumption have caused a high demand in using fertilizers, which are critical for plant growth and improving the crop yield. In general, conventional mineral fertilizers are soluble salts, which easily dissolve in the soil media for plant uptake. However, a large portion of these soluble salts leach to the water resources, resulting in eutrophication and nutrient loss in fertilization. Solid forms of insoluble fertilizers have also been applied, but the micronutrients are not so free to be easily bioavailable and easily transported to the water resources. However, when the plants are in need, these solid minerals are less effective in supplying micronutrients in

time due to their large size. To solve these problems, the application of nanoparticles could be helpful, hypothetically, in providing the micronutrients, minimizing the environmental contamination risks of soluble fertilizers, and the problem of less bioavailability of solid fertilizers, which is more environmentally benign [1,2]. As the dimensions of materials reduce from a large size below 100 nm, significant changes in characteristics can mainly occur due to an increase in relative surface area (per unit mass) and then an enhanced chemical reactivity [3].

In recent years, a large number of reports have analyzed the influence of various nanostructures on different crops, especially on their early developmental stage, seed germination, and seedling development. Seed germination (i.e., the emergence of the radicle and primary root's elongation) is considered as the most sensitive stage of a plant life cycle. Priming with nanoparticles (nano-priming) could lead to positive, negative, or no impacts on germination process, depending on NP type, size, concentration, duration of the exposure, or the growing conditions. Exact outcomes of seed priming are also considered as a technique to control the hydrate content of the seeds, stimulating metabolic activities for germination [4,5]. In priming with NPs, first, the NPs have to penetrate the sclereids barrier of the seed coat. Several studies have revealed that NPs reach the plant cells by crossing the intercellular spaces, binding to a carrier protein through aquaporin, ion channels, binding to organic materials or endocytosis by making new pores. The NP–plant interaction may result in morphological and physiological changes, depending on the characteristics of NPs [1,6]. It has also been confirmed that some metal-based NPs can cross the seed coat and stimulate the embryonic differentiation by inducing the enzymes that interrupt seed dormancy. NPs may translocate to the other plant parts and cause various structural or functional changes in those parts [7]. Previous reports have also suggested different biochemical mechanisms for the positive effect of nanomaterials upon NP exposure such as increased water uptake, remodeling of membrane lipids in seeds, enhanced sugar metabolism and energy production, and the stimulated antioxidant defense [5]. The impact of NPs on seed germination is related to their capability to enter the embryonic tissues through the seed coat. This capability is mostly related to the structure of the seed coat and differs upon each plant species, and the physical and chemical characteristics of the ambience [8].

There is also another reason for the evaluation of NP–plant interactions. Up to now, the amount of NPs that existed in the environment has been significantly lower than their toxic concentration. However, as NPs are now widely commercialized, the potential biological impacts of NPs should be carefully assessed. As NPs have the potential to find their pathway into the environment and plants continuously interact with soil, water, and air, these NPs may penetrate and translocate into the plant [9,10]. In this regard, there is a need to study the impact of NPs on plants.

Metals generally affect the seed germination process, biochemical and physiological profiles, and plant growth. Essential metals such as Zn, Mg, Cu, and Fe are crucial for living cells and their deficiency could lead to damage to the cell wall and DNA. Nevertheless, excessive amounts of these metals or the presence of non-essential metals (e.g., Ag) could be toxic due to causing oxidative stress, stimulating loss of membrane integrity, and injuring to proteins and DNA in a phytotoxic manner [11].

The objective of this study encompasses assessing the potential impact of seven metal-based nanoparticles that we biosynthesized, characterized, and applied in our previous works [12–14] on seedling and seedling growth of two popular plants of wheat and flax though a laboratory study. Calcinated zinc oxide (C-ZnO), non-calcinated zinc oxide (NC-ZnO), zinc (Zn), magnesium oxide (MgO), copper (Cu), and iron (Fe) include essential metals that are vital for plant growth. Ag NP was selected due to its extensive use in industry and it could reach the plants through surface water. Essential and nonessential elements might be absorbed by plants and according to their concentration, may result in toxicity. The effect of NPs have also been studied in some earlier reports. These nanoparticles showed both promoting and inhibition effects on plant growth [15–17]. Moreover,

the influence of metal salts, which are used as the precursors of the NPs during their biosynthesis, is compared with their counterpart NPs at the same concentrations. To the best of our knowledge, this report is the first to compare the positive and negative effects of these NPs on the seedling parameters.

## 2. Materials and Methods

The whole process is summarized in Figure 1.

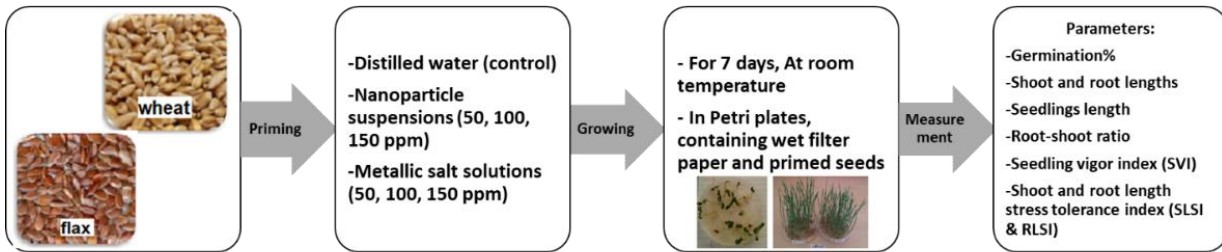

**Figure 1.** Graphical summary of the experiment. (ppm: part per million).

### 2.1. Biosynthesis of NPs

The plant extraction method, synthesis and characterization of applied NPs have been described in our previous works [12,14]. In summary, dried strawberry leaves were boiled in distilled water, filtered, and mixed with NP precursors. NPs were generated by a reduction of 0.01 M precursor salt solutions under heating, continuous stirring, and addition of the extract drop by drop. Produced NPs were washed with distilled water after centrifugation and dried at room temperature. C-ZnO and MgO NPs were calcinated in a furnace at 500 °C for 4 h. The biosynthesized NPs specified using different characterization techniques including UV–Vis spectroscopy, XRD, FESEM, EDS, photon cross-correlation spectroscopy (PCCS) and FTIR. Biosynthesized NPs, their counterpart precursors, their average sizes, and shapes are listed in Table 1.

**Table 1.** List of the applied biogenic nanoparticles (NPs), their counterpart precursors, the average size and shape of the NPs (nm: nanometer).

| Nanoparticle | C-ZnO | NC-ZnO | Zn | MgO | Ag | Cu | Fe |
|---|---|---|---|---|---|---|---|
| Precursor | $Zn(CH_2COO)_2$ | $Zn(CH_2COO)_2$ | $Zn(CH_2COO)_2$ | $MgSO_4$ | $AgNO_3$ | $CuSO_4$ | $FeCl_3$ |
| Average NP size (nm) | 40 | 25 | 100 | 65 | 50 | $180 \times 30$ | $130 \times 20$ |
| Shape | Spherical | spherical | sheets | Semi-spherical | spherical | sheets | sheets |

### 2.2. Preparation of Priming Solutions

Different concentrations (50, 100, 150 ppm) of NPs (C-ZnO, NC-ZnO, Zn, MgO, $AgNO_3$, Cu, and Fe) as well as their counterpart metal salts ($Zn(CH_2 COO)_2$, $MgSO_4$, $AgNO_3$, $CuSO_4$, and $FeCl_3$) prepared in distilled water and the NPs dispersed by ultrasonic vibrations for 20 min. Applied concentrations were selected considering previous reports [17]. All dilutions were freshly prepared before use.

### 2.3. Preparation of Seeds

Seeds of wheat (*Triticum aestivum* L.) variety Firuza 40 and flax (*Linum usitatissimum* L.) variety Semi Lini were used in this experiment. Each treatment consisted of 30 randomly selected seeds with three replications.

Seeds were checked visually to remove damaged seeds from the samples. Next, seeds were soaked in NP suspensions or metal salt solutions for 12 h. A set of seeds was used without providing any treatment as a control and soaked in distilled water.



### *2.4. In Vitro Germination of Seeds*

One piece of filter paper was placed into a Petri dish (10 cm in diameter), and to wet the paper, 5 mL distilled water was added using a Pasteur pipette. Then, 30 nano-primed seeds were transferred onto each filter paper. The Petri dishes were incubated at room temperature for seven days.

### *2.5. Measurement of Physiological Indexes*

Germination percentages, shoot length, root length, seedling length, root–shoot ratio, seedling vigor index (SVI), shoot length stress tolerance index (SLSI), and root length stress tolerance index (RLSI) were calculated on the second and seventh days. Means and standard deviations were derived from measurements on three replicates for each treatment and the controls. A seed was considered as germinated after the emergence of radicles or plumules from the seed coat [18,19].

(a)  Shoot and root length: On the second and seventh days of the experiment, 10 seedlings from Petri dish were randomly selected to measure the shoot and root lengths using a ruler with a centimeter and millimeter scale [20];
(b)  Seedling length = the sum of shoot length and root length of a seed [20];
(c)  Germination percentage (%) = (average number of germinated seeds/total number of seeds) × 100 [21];
(d)  Root/Shoot Ratio = average root length/average shoot length [21];
(e)  Seedling vigor index (SVI) = [average root length (cm) + average shoot length (cm)] × average germination percentage [22,23];
(f)  Shoot length stress tolerance index (SLSI%) = average shoot length of treated seedlings/ average shoot length of control seedlings × 100 [19,21]; and
(g)  Root length stress tolerance index (RLSI%) = average root length of treated seedlings/ average root length of control seedlings × 100 [19,21].

### *2.6. Statistical Analysis*

The obtained data were statistically analyzed using Microsoft Excel software (version 2019), SAS, and MSTAT-C statistical programs. Two-way analysis of variance (ANOVA) was applied to perform statistical analysis and a *p*-value < 0.05 considered as significant. Mean comparison was performed by the least significant different (LSD) test. Means and standard deviations obtained from measurements on three replicates for the control and each treatment.

## 3. Results

We observed that plant growth parameters of wheat and flax varied considerably among the plants and also different priming solutions with various concentrations. Moreover, both the effects of "stimulation" and "phytotoxicity", and in some cases, no significant effect of NPs and their counterpart salts was observed on germination and seedling growth.

### *3.1. Effect of Biogenic NPs and Their Counterpart Salts on Physiological Characteristics of Wheat Seedling*

Tables 2–5 summarize the effect of priming with biogenic NPs and their counterpart metal salts (precursors) on the seed germination parameters of wheat on the second and seventh days.

**Table 2.** Effect of biogenic NP priming on seed germination parameters of wheat on the second day under different nanoparticle concentrations. Data are presented as mean values ± SD for three independent experiments.

| Nanoparticle | Concentration (ppm) | Germination (%) | Shoot Length (cm) | Root Length (cm) | Seedling Length (cm) | Root to Shoot Ratio | SVI | SLSI (%) | RLSI (%) |
|---|---|---|---|---|---|---|---|---|---|
| C-ZNO | 0 (Control) | 83 ± 3 [e] | 2.6 ± 0.3 [b] | 4.3 ± 0.3 [ab] | 6.9 ± 0.4 [ab] | 1.6 ± 0.2 [c] | 575 ± 37 [c] | 100 ± 0 [b] | 100 ± 0 [b] |
| | 50 | 90 ± 2 [bc] | 2.4 ± 0.3 [c] | 3.1 ± 0.4 [c] | 5.5 ± 0.6 [d] | 1.3 ± 0.2 [d] | 493 ± 51 [de] | 90 ± 11 [cd] | 74 ± 12 [e] |
| | 100 | 86 ± 2 [cd] | 2.3 ± 0.3 [c] | 3.1 ± 0.3 [c] | 5.4 ± 0.5 [de] | 1.3 ± 0.2 [d] | 466 ± 44 [e] | 87 ± 8 [d] | 72 ± 7 [e] |
| | 150 | 93 ± 2 [b] | 2.7 ± 0.2 [b] | 4.3 ± 0.7 [ab] | 7.0 ± 0.8 [ab] | 1.6 ± 0.2 [c] | 657 ± 71 [b] | 102 ± 7 [b] | 102 ± 17 [b] |
| NC-ZnO | 0 (Control) | 83 ± 3 [e] | 2.6 ± 0.3 [b] | 4.3 ± 0.3 [ab] | 6.9 ± 0.4 [ab] | 1.6 ± 0.2 [c] | 575 ± 37 [c] | 100 ± 0 [b] | 100 ± 0 [b] |
| | 50 | 93 ± 2 [b] | 3.2 ± 0.2 [a] | 4.9 ± 0.5 [a] | 6.8 ± 0.7 [b] | 1.5 ± 0.1 [cd] | 750 ± 70 [a] | 120 ± 13 [b] | 115 ± 19 [a] |
| | 100 | 93 ± 2 [b] | 3.1 ± 0.2 [a] | 4.7 ± 0.6 [a] | 7.8 ± 0.7 [a] | 1.5 ± 0.1 [cd] | 726 ± 67 [a] | 117 ± 8 [a] | 110 ± 14 [a] |
| | 150 | 87 ± 1 [cd] | 2.8 ± 0.3 [ab] | 5.0 ± 0.4 [a] | 7.8 ± 0.6 [a] | 1.7 ± 0.1 [bc] | 675 ± 48 [b] | 107 ± 10 [b] | 116 ± 9 [a] |
| Zn | 0 (Control) | 83. ± 3 [e] | 2.6 ± 0.3 [b] | 4.3 ± 0.1 [ab] | 6.9 ± 0.4 [ab] | 1.6 ± 0.2 [c] | 574 ± 37 [c] | 100 ± 0 [b] | 100 ± 0 [b] |
| | 50 | 87 ± 3 [cd] | 2.7 ± 0.3 [b] | 4.5 ± 0.8 [ab] | 7.2 ± 0.9 [a] | 1.7 ± 0.3 [c] | 623 ± 80 [b] | 102 ± 13 [b] | 105 ± 19 [b] |
| | 100 | 97 ± 2 [ab] | 2.5 ± 0.2 [bc] | 4.4 ± 0.3 [ab] | 6.9 ± 0.2 [ab] | 1.8 ± 0.3 [bc] | 668 ± 18 [b] | 95 ± 9 [c] | 104 ± 8 [b] |
| | 150 | 99 ± 2 [a] | 2.7 ± 0.2 [b] | 4.6 ± 0.4 [ab] | 7.3 ± 0.5 [a] | 1.4 ± 0.2 [d] | 719 ± 46 [a] | 103 ± 8 [b] | 107 ± 9 [ab] |
| MgO | 0 (Control) | 83 ± 3 [e] | 2.6 ± 0.3 [b] | 4.3 ± 0.1 [ab] | 6.9 ± 0.4 [ab] | 1.6 ± 0.2 [c] | 574 ± 37 [c] | 100 ± 0 [b] | 100 ± 0 [b] |
| | 50 | 90 ± 2 [bc] | 2.4 ± 0.4 [bc] | 3.8 ± 0.3 [b] | 6.3 ± 0.7 [bc] | 1.6 ± 0.2 [c] | 565 ± 64 [cd] | 92 ± 14 [c] | 90 ± 8 [c] |
| | 100 | 93 ± 4 [b] | 2.5 ± 0.3 [bc] | 3.5 ± 0.2 [bc] | 5.9 ± 0.3 [c] | 1.4 ± 0.2 [d] | 552 ± 30 [cd] | 93 ± 11 [c] | 81 ± 5 [d] |
| | 150 | 93 ± 3 [b] | 2.7 ± 0.3 [b] | 3.8 ± 0.7 [b] | 6.5 ± 0.9 [bc] | 1.4 ± 0.2 [d] | 604 ± 87 [bc] | 100 ± 10 [b] | 90 ± 16 [cd] |
| Ag | 0 (Control) | 85 ± 3 [d] | 2.2 ± 0.3 [cd] | 3.8 ± 0.3 [b] | 6.0 ± 0.5 [c] | 1.8 ± 0.2 [bc] | 510 ± 42 [d] | 100 ± 0 [b] | 100 ± 0 [b] |
| | 50 | 87 ± 1 [cd] | 1.6 ± 0.4 [e] | 3.2 ± 0.3 [bc] | 4.8 ± 0.5 [f] | 2.1 ± 0.6 [ab] | 421 ± 40 [e] | 74 ± 19 [e] | 85 ± 7 [d] |
| | 100 | 90 ± 2 [bc] | 1.4 ± 0.4 [ef] | 3.3 ± 0.3 [bc] | 4.7 ± 0.2 [f] | 2.5 ± 0.9 [a] | 426 ± 22 [e] | 65 ± 17 [fg] | 87 ± 7 [d] |
| | 150 | 91 ± 2 [bc] | 1.5 ± 0.3 [ef] | 3.2 ± 0.5 [bc] | 4.7 ± 0.8 [f] | 2.2 ± 0.2 [ab] | 427 ± 71 [e] | 68 ± 15 [f] | 84 ± 12 [d] |
| Cu | 0 (Control) | 85 ± 3 [d] | 2.1 ± 0.3 [cd] | 3.8 ± 0.3 [b] | 6.0 ± 0.5 [c] | 1.8 ± 0.2 [bc] | 510 ± 4 [d] | 100 ± 0 [b] | 100 ± 0 [b] |
| | 50 | 92 ± 2 [b] | 2.2 ± 0.1 [c] | 3.5 ± 0.5 [bc] | 5.7 ± 0.6 [d] | 1.6 ± 0.1 [c] | 524 ± 5 [d] | 102 ± 5 [b] | 91 ± 13 [c] |
| | 100 | 87 ± 2 [cd] | 2.3 ± 0.2 [c] | 3.5 ± 0.5 [bc] | 5.9 ± 0.7 [cd] | 1.5 ± 0.1 [cd] | 510 ± 61 [d] | 108 ± 9 [b] | 92 ± 13 [c] |
| | 150 | 90 ± 2 [bc] | 2.0 ± 0.5 [d] | 3.7 ± 0.9 [b] | 5.7 ± 1.4 [cd] | 1.9 ± 0.1 [bc] | 516 ± 127 [d] | 92 ± 23 [c] | 97 ± 24 [bc] |
| Fe | 0 (Control) | 85 ± 3 [d] | 2.2 ± 0.3 [cd] | 3.8 ± 0.3 [b] | 6.0 ± 0.5 [c] | 1.8 ± 0.2 [bc] | 510 ± 42 [d] | 100 ± 0 [b] | 100 ± 0 [b] |
| | 50 | 90 ± 2 [bc] | 2.5 ± 0.4 [bc] | 3.1 ± 0.2 [c] | 5.6 ± 0.4 [d] | 1.3 ± 0.2 [de] | 504 ± 36 [d] | 115 ± 20 [a] | 81 ± 40 [d] |
| | 100 | 89 ± 3 [c] | 1.9 ± 0.5 [de] | 2.9 ± 1.1 [cd] | 4.8 ± 1.5 [f] | 1.5 ± 0.4 [cd] | 427 ± 136 [e] | 87 ± 23 [d] | 76 ± 29 [de] |
| | 150 | 90 ± 2 [bc] | 2.1 ± 0.3 [cd] | 3.4 ± 0.8 [bc] | 5.5 ± 1.0 [d] | 1.6 ± 0.3 [c] | 495 ± 91 [de] | 97 ± 12 [c] | 89 ± 21 [cd] |
| *p*-value | - | 0.0050 | 0.0011 | 0.0002 | 0.0059 | 0.0010 | 0.0009 | 0.0016 | 0.0007 |

The same letters within a column showed no significant difference at a 95% probability level at the $p < 0.05$ level. ppm: part per million, nm: nanometer, cm: centimeter, SD: standard deviation, *p*-value: probability value, SVI: seedling vigor index, SLSI: shoot length stress tolerance index, RLSI: root length stress tolerance index.

The data showed that exposure to different concentrations of biogenic NPs resulted in an increase in germination percentage (G%) of all the treatments over the control on the second day (Table 2). The maximum G% was related to the seeds primed with 150 ppm of Zn NP at 98%. Vice versa, a decrease in G% occurred in samples treated with metal salts compared to the control, except for the sample primed with 50 ppm of zinc acetate, which was similar to the control (Table 4).

On the seventh day, an increase was observed in most of the samples (Table 3). Interestingly, for most of the samples treated with metal salts, the G% values were similar or close to the control (Table 5). Among all of the applied NPs, Zn NPs were found to be more effective in developing the seed germination of wheat seeds.

**Table 3.** Effect of biogenic NP priming on seed germination parameters of wheat on the seventh day under different nanoparticle concentrations. Data are presented as mean values ± SD for three independent experiments. The same letters within a column showed no significant difference at a 95% probability level at the $p < 0.05$ level.

| NP | Concentration (ppm) | Germination (%) | Shoot Length (cm) | Root Length (cm) | Seedling Length (cm) | Root to Shoot Ratio | SVI | SLSI (%) | RLSI (%) |
|---|---|---|---|---|---|---|---|---|---|
| C-ZNO | 0 (Control) | 83 ± 3 [cd] | 11.2 ± 0.6 [bc] | 9.6 ± 0.5 [f] | 20.8 ± 0.9 [de] | 0.8 ± 0.0 [d] | 1736 ± 86 [g] | 100 ± 0 [b] | 100 ± 0 [c] |
| | 50 | 90 ± 1 [b] | 11.4 ± 0.5 [bc] | 11.4 ± 1.4 [d] | 22.7 ± 0.9 [cd] | 1.0 ± 0.1 [c] | 2047 ± 86 [d] | 101 ± 4 [b] | 118 ± 14 [bc] |
| | 100 | 90 ± 3 [b] | 11.1 ± 0.8 [c] | 10.5 ± 0.9 [e] | 22.5 ± 0.5 [cd] | 1.0 ± 0.2 [c] | 2026 ± 51 [d] | 98 ± 7 [bc] | 120 ± 10 [bc] |
| | 150 | 93 ± 2 [ab] | 11.2 ± 0.8 [bc] | 14.2 ± 0.9 [a] | 25.4 ± 0.7 [a] | 1.3 ± 0.2 [a] | 2368 ± 69 [a] | 99 ± 7 [bc] | 145 ± 10 [a] |
| NC-ZnO | 0 (Control) | 83 ± 3 [cd] | 11.2 ± 0.6 [bc] | 9.6 ± 0.5 [f] | 20.8 ± 0.9 [de] | 0.8 ± 0.0 [d] | 1736 ± 86 [g] | 100 ± 0 [b] | 100 ± 0 [c] |
| | 50 | 90 ± 3 [b] | 11.2 ± 1.0 [bc] | 13.4 ± 1.1 [b] | 24.6 ± 1.7 [ab] | 1.2 ± 0.1 [b] | 2226 ± 16 [b] | 100 ± 9 [b] | 141 ± 12 [a] |
| | 100 | 90 ± 2 [b] | 11.9 ± 0.6 [b] | 13.4 ± 0.7 [b] | 25.2 ± 0.6 [a] | 1.1 ± 0.1 [bc] | 2273 ± 58 [b] | 105 ± 5 [ab] | 139 ± 8 [a] |
| | 150 | 87 ± 3 [c] | 11.6 ± 0.4 [b] | 14.0 ± 1.8 [a] | 24.6 ± 1.7 [ab] | 1.2 ± 0.2 [b] | 2218 ± 15 [b] | 103 ± 4 [ab] | 144 ± 19 [a] |
| Zn | 0 (Control) | 83 ± 3 [cd] | 11.2 ± 0.6 [bc] | 9.6 ± 0.5 [f] | 20.8 ± 0.9 [de] | 0.8 ± 0.0 [d] | 1736 ± 86 [g] | 100 ± 0 [b] | 100 ± 0 [c] |
| | 50 | 96 ± 3 [a] | 12.2 ± 2.7 [a] | 12.4 ± 0.5 [c] | 24.6 ± 2.6 [ab] | 1.0 ± 0.2 [c] | 2380 ± 25 [a] | 109 ± 24 [a] | 129 ± 5 [ab] |
| | 100 | 87 ± 3 [c] | 11.4 ± 1.1 [bc] | 12.6 ± 0.5 [c] | 24.0 ± 1.2 [b] | 1.1 ± 0.1 [bc] | 2079 ± 10 [cd] | 101 ± 10 [b] | 132 ± 5 [ab] |
| | 150 | 97 ± 1 [a] | 11.7 ± 0.6 [b] | 12.7 ± 1.0 [bc] | 24.5 ± 1.6 [ab] | 1.1 ± 0.0 [c] | 2368 ± 15 [a] | 104 ± 6 [ab] | 133 ± 11 [ab] |
| MgO | 0 (Control) | 83 ± 3 [cd] | 11.2 ± 0.6 [bc] | 9.6 ± 0.5 [f] | 20.8 ± 0.9 [de] | 0.8 ± 0.0 [d] | 1736 ± 86 [g] | 100 ± 0 [b] | 100 ± 0 [c] |
| | 50 | 93 ± 2 [ab] | 11.5 ± 0.0 [bc] | 12.1 ± 0.8 [c] | 23.6 ± 0.8 [bc] | 1.0 ± 0.0 [c] | 2202 ± 80 [b] | 102 ± 0 [b] | 126 ± 9 [b] |
| | 100 | 93 ± 3 [ab] | 10.9 ± 0.5 [d] | 12.2 ± 0.5 [c] | 23.1 ± 0.5 [c] | 1.1 ± 0.1 [bc] | 2158 ± 44 [bc] | 96 ± 4 [bc] | 128 ± 5 [b] |
| | 150 | 93 ± 2 [ab] | 11.5 ± 0.57 [c] | 13.4 ± 1.4 [b] | 24.9 ± 1.9 [a] | 1.2 ± 0.1 [b] | 2321 ± 18 [a] | 102 ± 5 [b] | 139 ± 14 [a] |
| Ag | 0 (Control) | 89 ± 2 [bc] | 11.2 ± 1.1 [bc] | 8.3 ± 2.6 [g] | 19.6 ± 3.2 [e] | 0.7 ± 0.2 [e] | 1743 ± 28 [g] | 100 ± 0 [b] | 100 ± 0 [c] |
| | 50 | 89 ± 3 [bc] | 10.8 ± 0.9 [d] | 9.8 ± 1.6 [ef] | 20.6 ± 2.1 [de] | 0.9 ± 0.1 [d] | 1848 ± 18 [f] | 100 ± 8 [b] | 100 ± 16 [c] |
| | 100 | 91 ± 3 [b] | 9.9 ± 0.9 [e] | 7.8 ± 1.1 [gh] | 17.8 ± 1.8 [f] | 0.8 ± 0.1 [d] | 1617 ± 16 [h] | 92 ± 8 [c] | 80 ± 11 [d] |
| | 150 | 92 ± 3 [ab] | 9.4 ± 0.8 [f] | 6.2 ± 1.0 [i] | 15.6 ± 1.6 [g] | 0.6 ± 0.1 [ef] | 1438 ± 149 [i] | 86 ± 8 [cd] | 63 ± 10 [e] |
| Cu | 0 (Control) | 89 ± 2 [bc] | 11.2 ± 1.1 [bc] | 8.3 ± 2.6 [g] | 19.6 ± 3.2 [e] | 0.7 ± 0.2 [e] | 1743 ± 283 [g] | 100 ± 0 [b] | 100 ± 0 [c] |
| | 50 | 95 ± 2 [a] | 12.2 ± 0.8 [a] | 10.9 ± 1.1 [de] | 23.1 ± 1.3 [c] | 0.9 ± 0.1 [d] | 2052 ± 11 [d] | 112 ± 7 [a] | 111 ± 11 [bc] |
| | 100 | 90 ± 3 [bc] | 11.4 ± 1.0 [bc] | 10.6 ± 1.5 [e] | 22.0 ± 2.0 [d] | 0.9 ± 0.1 [d] | 1976 ± 18 [de] | 105 ± 10 [ab] | 108 ± 16 [c] |
| | 150 | 91 ± 1 [b] | 11.9 ± 0.9 [ab] | 12.4 ± 1.1 [b] | 24.4 ± 1.7 [ab] | 1.0 ± 0.1 [c] | 2219 ± 15 [b] | 110 ± 8 [a] | 126 ± 11 [b] |
| Fe | 0 (Control) | 89 ± 2 [c] | 11.2 ± 1.1 [bc] | 8.3 ± 2.6 [g] | 19.6 ± 3.2 [e] | 0.7 ± 0.2 [e] | 1743 ± 28 [g] | 100 ± 0 [b] | 100 ± 0 [c] |
| | 50 | 91 ± 2 [b] | 12.0 ± 0.6 [ab] | 11.1 ± 0.8 [d] | 23.1 ± 1.1 [c] | 0.9 ± 0.1 [cd] | 2112 ± 10 [c] | 111 ± 5 [a] | 113 ± 8 [bc] |
| | 100 | 92 ± 3 [b] | 11.0 ± 0.8 [cd] | 10.7 ± 1.1 [de] | 21.6 ± 1.6 [d] | 1.0 ± 0.1 [cd] | 1987 ± 14 [de] | 101 ± 7 [b] | 109 ± 11 [c] |
| | 150 | 91 ± 1 [b] | 11.3 ± 1.4 [bc] | 10.6 ± 1.0 [de] | 21.9 ± 1.6 [d] | 0.9 ± 0.1 [cd] | 2002 ± 14 [d] | 104 ± 13 [ab] | 108 ± 11 [c] |
| *p*-value | | 0.0100 | 0.0008 | 0.0066 | 0.0003 | 0.0040 | 0.0009 | 0.0160 | 0.0038 |

The same letters within a column showed no significant difference at a 95% probability level at the $p < 0.05$ level. ppm: part per million, nm: nanometer, cm: centimeter, SD: standard deviation, *p*-value: probability value, SVI: seedling vigor index, SLSI: shoot length stress tolerance index, RLSI: root length stress tolerance index.

On the second day of the experiment, the best results of shoot elongation were related to the seeds treated with NC-ZnO NPs among all of the used NPs (Table 2). In the case of priming with salts (Table 4), the SLSI values were less than the control, except for 100 ppm zinc acetate, which was close to the control. Ag NPs induced a significant decrease in shoot length and SLSI, both on the second and seventh days. According to Tables 3 and 5, the maximum SLSI on the seventh day are related to the concentrations of 50 ppm of Zn and Fe, and also 50 and 150 ppm of Cu NPs (108 to 112%). In salt primed samples, only a priming with 100 ppm of FeCl₃ had a positive effect on SLSI (125%) and no other significant increase was observed. There were also a decrease in shoot length in the CuSO₄ primed samples.

According to Tables 2 and 4, on the second day of the test, the most significant promoting effect on root length and RLSI was related to the soaking with NC-ZnO NPs. Root lengths of the Zn NPs treated seeds were similar to the control and the other NP

treatments had an inverse effect on root elongation. Considering the results of the priming with salts, both an increase and decrease in root lengths and RLSI were observed.

**Table 4.** Effect of metal salt (precursors) priming on seed germination parameters of wheat on the second day under different nanoparticle concentrations. Data are presented as mean values $\pm$ SD for three independent experiments. The same letters within a column showed no significant difference at a 95% probability level at the *p* < 0.05 level.

| Metal Salt | Concentration (ppm) | Germination (%) | Shoot Length (cm) | Root Length (cm) | Seedling Length (cm) | Root to Shoot Ratio | SVI | SLSI (%) | RLSI (%) |
|---|---|---|---|---|---|---|---|---|---|
| Zn(CH₃CO₂)₂ | 0 (Control) | 99 ± 2 ᵃ | 1.8 ± 0.6 ᵃ | 2.4 ± 0.4 ᶜᵈ | 4.2 ± 0.7 ᵇᶜ | 1.5 ± 0.4 ᵈᵉ | 415 ± 6 ᵃᵇ | 100 ± 0 ᵃ | 100 ± 0 ᶜ |
| | 50 | 99 ± 2 ᵃ | 1.7 ± 0.2 ᵃ | 3.0 ± 0.3 ᵇ | 4.7 ± 0.3 ᵇ | 1.8 ± 0.3 ᶜᵈ | 464 ± 35 ᵃ | 95 ± 11 ᵃᵇ | 124 ± 14 ᵃᵇ |
| | 100 | 93 ± 3 ᶜ | 1.8 ± 0.2 ᵃ | 3.3 ± 0.3 ᵃ | 5.1 ± 0.4 ᵃ | 1.9 ± 0.2 ᶜ | 467 ± 36 ᵃ | 100 ± 12 ᵃ | 139 ± 11 ᵃ |
| | 150 | 97 ± 3 ᵇ | 1.5 ± 0.1 ᵇ | 2.8 ± 0.5 ᶜ | 4.3 ± 0.5 ᵇᶜ | 1.9 ± 0.41 ᶜ | 412 ± 5 ᵃᵇ | 83 ± 5 ᶜ | 115 ± 22 ᵇ |
| MgSO₄ | 0 (Control) | 99 ± 2 ᵃ | 1.8 ± 0.6 ᵃ | 2.4 ± 0.4 ᶜᵈ | 4.2 ± 0.7 ᵇᶜ | 1.5 ± 0.4 ᵈᵉ | 415 ± 6 ᵃᵇ | 100 ± 0 ᵃ | 100 ± 0 ᶜ |
| | 50 | 87 ± 3 ᵉ | 1.2 ± 0.2 ᵈ | 1.5 ± 0.1 ᵉ | 2.8 ± 0.2 ᵉᶠ | 1.2 ± 0.1 ᶠ | 242 ± 19 ᵈ | 70 ± 9 ᵈ | 64 ± 4 ᶠ |
| | 100 | 93 ± 3 ᶜ | 1.6 ± 0.2 ᵃᵇ | 2.7 ± 0.3 ᶜ | 4.3 ± 0.2 ᵇᶜ | 1.8 ± 0.4 ᶜᵈ | 404 ± 12 ᵇ | 89 ± 13 ᵇᶜ | 114 ± 12 ᵇ |
| | 150 | 97 ± 2 ᵇ | 1.5 ± 0.2 ᵇ | 3.2 ± 0.2 ᵃᵇ | 4.7 ± 0.4 ᵇ | 2.1 ± 0.2 ᵇ | 452 ± 36 ᵃ | 85 ± 9 ᵇᶜ | 131 ± 11 ᵃ |
| AgNO₃ | 0 (Control) | 99 ± 2 ᵃ | 1.8 ± 0.6 ᵃ | 2.4 ± 0.4 ᶜᵈ | 4.2 ± 0.7 ᵇᶜ | 1.5 ± 0.405 ᵈᵉ | 415 ± 6 ᵃᵇ | 100 ± 0 ᵃ | 100 ± 0 ᶜ |
| | 50 | 83 ± 3 ᶠ | 1.4 ± 0.2 ᵇᶜ | 2.9 ± 0.2 ᵇᶜ | 4.3 ± 0.2 ᵇᶜ | 2.2 ± 0.4 ᵃᵇ | 358 ± 20 ᶜ | 78 ± 11 ᶜᵈ | 121 ± 8 ᵃᵇ |
| | 100 | 90 ± 3 ᵈ | 1.1 ± 0.1 ᵈ | 2.4 ± 0.4 ᶜᵈ | 3.5 ± 0.3 ᵈ | 2.2 ± 0.5 ᵃ | 312 ± 31 ᶜᵈ | 61 ± 7 ᵉ | 99 ± 15 ᶜᵈ |
| | 150 | 97 ± 3 ᵇ | 1.3 ± 0.2 ᵇᶜ | 2.8 ± 0.3 ᶜ | 4.1 ± 0.4 ᶜᶜ | 2.2 ± 0.4 ᵃᵇ | 396 ± 40 ᵇ | 74 ± 12 ᵈ | 115 ± 14 ᵇ |
| CuSO₄ | 0 (Control) | 99 ± 2 ᵃ | 1.8 ± 0.6 ᵃ | 2.4 ± 0.4 ᶜᵈ | 4.2 ± 0.7 ᵇᶜ | 1.5 ± 0.4 ᵈᵉ | 415 ± 6 ᵃᵇ | 100 ± 0 ᵃ | 100 ± 0 ᶜ |
| | 50 | 91 ± 3 ᵈ | 1.5 ± 0.1 ᵇᶜ | 3.1 ± 0.4 ᵇ | 4.6 ± 0.4 ᵇ | 2.0 ± 0.3 ᵇ | 422 ± 3 ᵃᵇ | 86 ± 4 ᵇᶜ | 129 ± 15 ᵃᵇ |
| | 100 | 90 ± 4 ᵈ | 1.3 ± 0.3 ᵇᶜ | 1.6 ± 0.5 ᵉ | 2.9 ± 0.7 ᵉ | 1.3 ± 0.3 ᶠ | 266 ± 60 ᵈ | 74 ± 16 ᵈ | 68 ± 20 ᵉᶠ |
| | 150 | 97 ± 3 ᵇ | 1.2 ± 0.2 ᵈ | 1.4 ± 0.2 ᶠ | 2.6 ± 0.2 ᶠᵍ | 1.2 ± 0.3 ᶠ | 256 ± 1 ᵈ | 67 ± 10 ᵈᵉ | 60 ± 8 ᶠ |
| FeCl₃ | 0 (Control) | 99 ± 2 ᵃ | 1.8 ± 0.6 ᵃ | 2.4 ± 0.4 ᶜᵈ | 4.2 ± 0.7 ᶜᵈ | 1.5 ± 0.4 ᵈᵉ | 415 ± 6 ᶜᵈ | 100 ± 0 ᵃ | 100 ± 0 ᶜ |
| | 50 | 91 ± 3 ᵈ | 1.6 ± 0.2 ᵃ | 1.7 ± 0.1 ᵉ | 3.3 ± 0.1 ᵈ | 1.2 ± 0.1 ᶠ | 299 ± 9 ᵈ | 91 ± 6 ᵇ | 70 ± 2 ᵉᶠ |
| | 100 | 95 ± 3 ᶜ | 1.4 ± 0.4 ᵇ | 1.6 ± 0.5 ᵉ | 3.0 ± 0.6 ᵉ | 1.6 ± 0.4 ᵈ | 283 ± 57 ᵈ | 79 ± 14 ᶜ | 77 ± 12 ᵉ |
| | 150 | 98 ± 3 ᵃ | 1.4 ± 0.5 ᵇ | 2.1 ± 0.5 ᵈ | 3.5 ± 0.5 ᵈ | 1.8 ± 0.7 ᶜ | 346 ± 49 ᶜ | 81 ± 17 ᶜ | 86 ± 11 ᵈ |
| *p*-value | | 0.0008 | 0.0111 | 0.0101 | 0.0022 | 0.0009 | 0.0100 | 0.0061 | 0.0004 |

The same letters within a column showed no significant difference at a 95% probability level at the *p* < 0.05 level. ppm: part per million, nm: nanometer, cm: centimeter, SD: standard deviation, *p*-value: probability value, SVI: seedling vigor index, SLSI: shoot length stress tolerance index, RLSI: root length stress tolerance index.

On the seventh day (Tables 3 and 5), there was a significant improvement in root length for all of the NP-primed seeds, except for the Ag NP-primed samples. NC-ZnO had the best effect on root length development and Ag NP showed a dose dependent inhibition effect on root lengths. For metal salt priming cases, the best results in root elongation and RLSI were related to FeCl₃ and then zinc acetate priming. For the other samples, there was a decrease in root length, mainly in the CuSO₄ primed seeds.

Under specific conditions, a higher proportion of roots can help plants to compete more efficiently for water uptake and soil resources, while a higher proportion of shoots can help plants to collect more light energy [24]. To evaluate the effect of metal NPs or metal salts on seedling growth, the seedling vigor index (SVI) could be used as a phytotoxicity index [25].

**Table 5.** Effect of metal salt (precursors) priming on seed germination parameters of wheat on the seventh day, under different nanoparticle concentrations. Data are presented as mean values $\pm$ SD for three independent experiments. The same letters within a column showed no significant difference at a 95% probability level at the $p < 0.05$ level.

| Metal Salt | Concentration (ppm) | Germination (%) | Shoot Length (cm) | Root Length (cm) | Seedling Length (cm) | Root to Shoot Ratio | SVI | SLSI (%) | RLSI (%) |
|---|---|---|---|---|---|---|---|---|---|
| Zn (CH$_3$ CO$_2$)$_2$ | 0 (Control) | 99 $\pm$ 2 [a] | 10.0 $\pm$ 0.7 [b] | 9.0 $\pm$ 0.8 [c] | 19.0 $\pm$ 1.4 [d] | 0.9 $\pm$ 0.1 [b] | 1881 $\pm$ 141 [c] | 100 $\pm$ 0 [b] | 100 $\pm$ 0 [b] |
| | 50 | 98 $\pm$ 2 [a] | 10.0 $\pm$ 1.5 [b] | 8.3 $\pm$ 1.7 [cd] | 18.3 $\pm$ 2.4 [e] | 0.8 $\pm$ 0.2 [c] | 1802 $\pm$ 237 [c] | 99 $\pm$ 14 [b] | 92 $\pm$ 19 [c] |
| | 100 | 98 $\pm$ 3 [a] | 10.3 $\pm$ 0.8 [b] | 10.6 $\pm$ 1.3 [b] | 20.9 $\pm$ 2.0 [b] | 1.0 $\pm$ 0.1 [a] | 2043 $\pm$ 192 [a] | 103 $\pm$ 8 [ab] | 115 $\pm$ 9 [a] |
| | 150 | 99 $\pm$ 2 [a] | 9.8 $\pm$ 1.2 [bc] | 10.6 $\pm$ 1.6 [b] | 20.4 $\pm$ 2.4 [bc] | 1.1 $\pm$ 0.1 [a] | 2020 $\pm$ 242 [a] | 98 $\pm$ 12 [b] | 118 $\pm$ 18 [a] |
| MgSO$_4$ | 0 (Control) | 99 $\pm$ 2 [a] | 10.0 $\pm$ 0.7 [b] | 9.0 $\pm$ 0.8 [c] | 19.0 $\pm$ 1.4 [d] | 0.9 $\pm$ 0.1 [b] | 1881 $\pm$ 112 [c] | 100 $\pm$ 0 [b] | 100 $\pm$ 0 [b] |
| | 50 | 95 $\pm$ 3 [bc] | 8.6 $\pm$ 2.6 [cd] | 7.4 $\pm$ 1.8 [d] | 16.1 $\pm$ 4.3 [f] | 0.9 $\pm$ 0.2 [b] | 1526 $\pm$ 409 [e] | 86 $\pm$ 26 [cd] | 83 $\pm$ 20 [de] |
| | 100 | 98 $\pm$ 2 [a] | 10.1 $\pm$ 0.6 [b] | 8.7 $\pm$ 1.4 [cd] | 18.8 $\pm$ 1.5 [de] | 0.9 $\pm$ 0.1 [bc] | 1843 $\pm$ 155 [c] | 100 $\pm$ 6 [b] | 98 $\pm$ 16 [b] |
| | 150 | 99 $\pm$ 2 [a] | 9.7 $\pm$ 1.1 [c] | 8.5 $\pm$ 1.9 [cd] | 18.2 $\pm$ 2.9 [e] | 0.9 $\pm$ 0.1 [bc] | 1639 $\pm$ 257 [d] | 97 $\pm$ 11 [bc] | 94 $\pm$ 21 [bc] |
| AgNO$_3$ | 0 (Control) | 99 $\pm$ 2 [a] | 10.0 $\pm$ 0.7 [b] | 9.0 $\pm$ 0.8 [c] | 19.0 $\pm$ 1.4 [d] | 0.9 $\pm$ 0.0 [b] | 1881 $\pm$ 121 [c] | 100 $\pm$ 0 [b] | 100 $\pm$ 0 [b] |
| | 50 | 97 $\pm$ 3 [ab] | 9.3 $\pm$ 0.7 [c] | 7.1 $\pm$ 1.4 [d] | 16.4 $\pm$ 2.0 [f] | 0.8 $\pm$ 0.1 [d] | 1591 $\pm$ 199 [de] | 92 $\pm$ 7 [c] | 79 $\pm$ 16 [de] |
| | 100 | 97 $\pm$ 3 [ab] | 7.7 $\pm$ 1.1 [d] | 4.6 $\pm$ 1.0 [e] | 12.3 $\pm$ 1.4 [g] | 0.6 $\pm$ 0.1 [e] | 1196 $\pm$ 140 [ef] | 77 $\pm$ 11 [d] | 51 $\pm$ 10 [f] |
| | 150 | 99 $\pm$ 2 [a] | 8.5 $\pm$ 1.0 [cd] | 4.2 $\pm$ 1.1 [e] | 12.7 $\pm$ 1.9 [g] | 0.5 $\pm$ 0.1 [f] | 1258 $\pm$ 186 [e] | 84 $\pm$ 10 [cd] | 47 $\pm$ 13 [fg] |
| CuSO$_4$ | 0 (Control) | 99 $\pm$ 2 [a] | 10.0 $\pm$ 0.7 [b] | 9.0 $\pm$ 0.8 [c] | 19.0 $\pm$ 1.4 [d] | 0.9 $\pm$ 0.0 [b] | 1881 $\pm$ 141 [c] | 100 $\pm$ 0 [b] | 100 $\pm$ 0 [b] |
| | 50 | 97 $\pm$ 3 [ab] | 7.7 $\pm$ 2.8 [d] | 4.5 $\pm$ 2.2 [e] | 12.3 $\pm$ 4.7 [g] | 0.6 $\pm$ 0.2 [e] | 1191 $\pm$ 457 [ef] | 77 $\pm$ 28 [d] | 50 $\pm$ 24 [f] |
| | 100 | 97 $\pm$ 3 [b] | 7.8 $\pm$ 1.3 [d] | 1.9 $\pm$ 0.7 [f] | 9.7 $\pm$ 1.9 [h] | 0.2 $\pm$ 0.1 [g] | 938 $\pm$ 182 [g] | 78 $\pm$ 13 [d] | 21 $\pm$ 8 [g] |
| | 150 | 99 $\pm$ 2 [a] | 5.5 $\pm$ 0.8 [e] | 1.2 $\pm$ 0.6 [f] | 6.8 $\pm$ 0.8 [i] | 0.2 $\pm$ 0.1 [g] | 670 $\pm$ 86 [h] | 55 $\pm$ 8 [e] | 14 $\pm$ 7 [h] |
| FeCl$_3$ | 0 (Control) | 99 $\pm$ 2 [a] | 10.0 $\pm$ 0.7 [b] | 9.0 $\pm$ 0.8 [c] | 19.0 $\pm$ 1.4 [d] | 0.9 $\pm$ 0.0 [b] | 1881 $\pm$ 141 [c] | 100 $\pm$ 0 [b] | 100 $\pm$ 0 [b] |
| | 50 | 87 $\pm$ 3 [d] | 11.1 $\pm$ 0.2 [a] | 8.2 $\pm$ 2.2 [cd] | 19.4 $\pm$ 1.9 [d] | 0.7 $\pm$ 0.2 [d] | 1679 $\pm$ 167 [d] | 99 $\pm$ 2 [b] | 86 $\pm$ 23 [d] |
| | 100 | 90 $\pm$ 3 [cd] | 11.1 $\pm$ 0.3 [a] | 10.7 $\pm$ 1.8 [b] | 21.9 $\pm$ 1.8 [b] | 1.0 $\pm$ 0.2 [ab] | 1971 $\pm$ 16 [ab] | 125 $\pm$ 3 [a] | 112 $\pm$ 20 [ab] |
| | 150 | 93 $\pm$ 3 [c] | 11.1 $\pm$ 0.5 [a] | 11.2 $\pm$ 0.6 [a] | 22.4 $\pm$ 0.6 [a] | 1.0 $\pm$ 0.1 a | 2088 $\pm$ 58 [a] | 99 $\pm$ 4 [b] | 117 $\pm$ 7 [a] |
| *p*-value | | 0.0200 | 0.0000 | 0.0055 | 0.0007 | 0.0108 | 0.0077 | 0.0100 | 0.0006 |

The same letters within a column showed no significant difference at a 95% probability level at the $p < 0.05$ level. ppm: part per million, nm: nanometer, cm: centimeter, SD: standard deviation, *p*-value: probability value, SVI: seedling vigor index, SLSI: shoot length stress tolerance index, RLSI: root length stress tolerance index.

### 3.2. Effect of Biogenic NPs and Their Counterpart Salts on Physiological Characteristics of Flax Seedling

Tables 6–9 summarize the effect of priming with biogenic NPs and their counterpart metal salts (precursors) on seed germination parameters of flax on the second and seventh days.

Considering the results of Tables 6 and 8, in the early stages of the seedling, the maximum seed germination percentage (90%) was related to the flax seeds soaked with 100 ppm C-ZnO NPs and all of the other treatments showed G% similar or less than the control. The minimum G% was related to the seeds soaked with 100 ppm of Cu NPs. Among the metal salts, zinc acetate had an improving effect on G%. On the seventh day of the experiment (Tables 7 and 9), only 150 ppm of the C-ZnO NP suspension had an improving effect over the control and all of the other NP treatments showed an inhibition effect. Among the salts, G% of the 150 ppm of the MgSO$_4$ treated seeds was close to the control and the others were less than the control. A total of 150 ppm of AgNO$_3$ had such a severe toxic effect that no gemination was observed.

According to Tables 6 and 8, on the second day of the test, all of the NP treatments showed an enhancement in shoot length over the control. The most effective NP was NC-ZnO and the less effective one was Ag. The most effective salts in improving shoot length were MgSO$_4$ and then zinc acetate. Tables 7 and 9 record the results of the seventh day of the experiment, which showed that the most effective NP in shoot length development of flax seeds was Ag. MgSO$_4$ and FeCl$_3$ were the most effective in shoot length development.

**Table 6.** Effect of biogenic NP priming on seed germination parameters of flax on the second day under different nanoparticle concentrations. Data are presented as mean values $\pm$ SD for three independent experiments. The same letters within a column showed no significant difference at a 95% probability level at the $p < 0.05$ level.

| NP | Concentration (ppm) | Germination (%) | Shoot Length (cm) | Root Length (cm) | Seedling Length (cm) | Root to Shoot Ratio | SVI | SLSI (%) | RLSI (%) |
|---|---|---|---|---|---|---|---|---|---|
| Non (Control) | 0 | 87 $\pm$ 2 [b] | 0.3 $\pm$ 0.0 [ef] | 2.2 $\pm$ 0.3 [cd] | 2.6 $\pm$ 0.3 [c] | 7.1 $\pm$ 1.3 [a] | 222 $\pm$ 25 [c] | 100 $\pm$ 0 [f] | 100 $\pm$ 0 [bc] |
| C-ZnO | 50 | 86 $\pm$ 2 [b] | 0.5 $\pm$ 0.0 [de] | 1.8 $\pm$ 0.5 [e] | 2.3 $\pm$ 0.5 [cd] | 3.8 $\pm$ 0.9 [d] | 201 $\pm$ 43 [cd] | 150 $\pm$ 14 [e] | 82 $\pm$ 21 [e] |
| | 100 | 90 $\pm$ 2 [a] | 0.7 $\pm$ 0.1 [bc] | 2.5 $\pm$ 0.2 [c] | 3.2 $\pm$ 0.2 [b] | 3.9 $\pm$ 0.7 [d] | 286 $\pm$ 23 [b] | 206 $\pm$ 36 [b] | 112 $\pm$ 9 [b] |
| | 150 | 83 $\pm$ 3 [bc] | 0.7 $\pm$ 0.2 [b] | 3.1 $\pm$ 0.4 [a] | 3.9 $\pm$ 0.5 [a] | 4.3 $\pm$ 1.0 [cd] | 322 $\pm$ 41 [a] | 231 $\pm$ 53 [ab] | 139 $\pm$ 18 [a] |
| NC-ZnO | 50 | 80 $\pm$ 3 [c] | 0.6 $\pm$ 0.2 [c] | 3.0 $\pm$ 0.4 [bc] | 3.4 $\pm$ 0.2 [b] | 4.7 $\pm$ 2.0 [c] | 270 $\pm$ 21 [b] | 200 $\pm$ 57 [b] | 122 $\pm$ 17 [ab] |
| | 100 | 80 $\pm$ 3 [c] | 0.8 $\pm$ 0.1 [a] | 2.3 $\pm$ 0.1 [cd] | 3.2 $\pm$ 0.2 [b] | 2.9 $\pm$ 0.5 [e] | 256 $\pm$ 18 [bc] | 262 $\pm$ 47 [a] | 105 $\pm$ 7 [b] |
| | 150 | 83 $\pm$ 3 [c] | 0.8 $\pm$ 0.1 [a] | 2.2 $\pm$ 0.2 [cd] | 3.0 $\pm$ 0.2 [b] | 2.7 $\pm$ 0.5 [e] | 252 $\pm$ 16 [bc] | 256 $\pm$ 34 [a] | 98 $\pm$ 10 [bc] |
| Zn | 50 | 87 $\pm$ 2 [b] | 0.7 $\pm$ 0.2 [b] | 3.0 $\pm$ 0.5 [ab] | 3.7 $\pm$ 0.7 [a] | 4.3 $\pm$ 1.2 [bc] | 324 $\pm$ 59 [a] | 231 $\pm$ 78 [ab] | 134 $\pm$ 22 [a] |
| | 100 | 73 $\pm$ 4 [d] | 0.7 $\pm$ 0.1 [b] | 2.5 $\pm$ 0.6 [c] | 3.2 $\pm$ 0.5 [b] | 3.8 $\pm$ 1.5 [d] | 235 $\pm$ 38 [c] | 219 $\pm$ 38 [b] | 112 $\pm$ 27 [b] |
| | 150 | 73 $\pm$ 4 [d] | 0.7 $\pm$ 0.2 [b] | 3.1 $\pm$ 0.4 [ab] | 3.8 $\pm$ 0.6 [a] | 4.6 $\pm$ 1.0 [c] | 277 $\pm$ 43 [b] | 219 $\pm$ 66 [b] | 137 $\pm$ 18 [a] |
| MgO | 50 | 77 $\pm$ 3 [d] | 0.4 $\pm$ 0.1 [de] | 2.3 $\pm$ 0.6 [cd] | 2.7 $\pm$ 0.7 [c] | 5.5 $\pm$ 1.7 [b] | 210 $\pm$ 55 [cd] | 137 $\pm$ 42 [de] | 103 $\pm$ 28 [bc] |
| | 100 | 80 $\pm$ 2 [c] | 0.6 $\pm$ 0.1 [d] | 2.3 $\pm$ 0.3 [cd] | 2.9 $\pm$ 0.2 [bc] | 4.2 $\pm$ 0.9 [c] | 229 $\pm$ 20 [c] | 175 $\pm$ 27 [ce] | 103 $\pm$ 12 [bc] |
| | 150 | 73 $\pm$ 3 [d] | 0.7 $\pm$ 0.1 [bc] | 2.4 $\pm$ 0.4 [c] | 3.1 $\pm$ 0.4 [b] | 3.7 $\pm$ 0.8 [d] | 226 $\pm$ 27 [c] | 206 $\pm$ 28 [b] | 108 $\pm$ 16 [b] |
| Ag | 50 | 83 $\pm$ 3 [c] | 0.4 $\pm$ 0.1 [ef] | 1.6 $\pm$ 0.6 [ef] | 2.0 $\pm$ 0.7 [de] | 4.6 $\pm$ 1.3 [c] | 167 $\pm$ 63 [e] | 114 $\pm$ 48 [f] | 73 $\pm$ 29 [f] |
| | 100 | 84 $\pm$ 3 [c] | 0.4 $\pm$ 0.1 [de] | 1.7 $\pm$ 0.2 [ef] | 2.1 $\pm$ 0.1 [d] | 4.5 $\pm$ 3.1 [c] | 175 $\pm$ 8 [de] | 137 $\pm$ 42 [ef] | 74 $\pm$ 9 [f] |
| | 150 | 80 $\pm$ 3 [c] | 0.5 $\pm$ 0.6 [ce] | 1.4 $\pm$ 0.3 [fg] | 1.9 $\pm$ 0.4 [f] | 3.3 $\pm$ 1.4 [de] | 151 $\pm$ 32 [ef] | 144 $\pm$ 52 [e] | 63 $\pm$ 16 [f] |
| Cu | 50 | 87 $\pm$ 2 [b] | 0.6 $\pm$ 0.2 [cd] | 2.3 $\pm$ 0.5 [cd] | 2.9 $\pm$ 0.6 [bc] | 4.2 $\pm$ 0.7 [c] | 254 $\pm$ 53 [bc] | 181 $\pm$ 51 [cd] | 105 $\pm$ 21 [bc] |
| | 100 | 61 $\pm$ 4 [e] | 0.4 $\pm$ 0.2 [e] | 1.5 $\pm$ 0.4 [f] | 1.9 $\pm$ 0.5 [f] | 4.5 $\pm$ 2.4 [c] | 115 $\pm$ 28 [g] | 118 $\pm$ 51 [f] | 67 $\pm$ 17 [f] |
| | 150 | 76 $\pm$ 4 [d] | 0.6 $\pm$ 0.1 [c] | 1.6 $\pm$ 0.4 [f] | 2.2 $\pm$ 0.4 [cd] | 2.6 $\pm$ 0.9 [ef] | 167 $\pm$ 33 [e] | 194 $\pm$ 40 [bc] | 70 $\pm$ 18 [f] |
| Fe | 50 | 86 $\pm$ 2 [b] | 0.4 $\pm$ 0.2 [e] | 1.3 $\pm$ 0.3 [h] | 1.9 $\pm$ 0.3 [f] | 3.4 $\pm$ 1.3 [d] | 145 $\pm$ 25 [ef] | 131 $\pm$ 56 [de] | 56 $\pm$ 12 [g] |
| | 100 | 77 $\pm$ 4 [d] | 0.5 $\pm$ 0.1 [d] | 1.8 $\pm$ 0.3 [e] | 2.3 $\pm$ 0.3 [cd] | 3.9 $\pm$ 1.4 [cd] | 179 $\pm$ 22 [de] | 156 $\pm$ 44 [e] | 82 $\pm$ 14 [e] |
| | 150 | 87 $\pm$ 2 [b] | 0.6 $\pm$ 0.1 [c] | 1.6 $\pm$ 0.5 [f] | 2.2 $\pm$ 0.1 [cd] | 2.5 $\pm$ 0.5 [ef] | 190 $\pm$ 53 [d] | 194 $\pm$ 41 [bc] | 70 $\pm$ 23 [f] |
| *p*-value | | 0.0011 | 0.0001 | 0.0300 | 0.0005 | 0.0066 | 0.0110 | 0.0009 | 0.0043 |

The same letters within a column showed no significant difference at a 95% probability level at the $p < 0.05$ level. ppm: part per million, nm: nanometer, cm: centimeter, SD: standard deviation, *p*-value: probability value, SVI: seedling vigor index, SLSI: shoot length stress tolerance index, RLSI: root length stress tolerance index.

On the second day (Tables 6 and 8), Zn NPs had the best effect on root growth. Ag and Fe NPs had the best inhibition effect. $MgSO_4$ was the most effective salt in root length development and $CuSO_4$ was the most toxic. On the seventh day of the experiment, a significant increase in root length was observed in most of the treatments, especially in the Zn NPs treated samples and the root lengths were about two to three times higher than the control, which is considered as a great positive effect. Cu of 100 ppm besides 50 and 150 ppm concentrations of Fe NPs showed a notable toxic effect on root length parameter. Zinc acetate and $MgSO_4$ solutions induced a significant root length increase of about twofold over the control. A total of 100 and 150 ppm of $CuSO_4$ priming inhibited the root growth to the lengths to about 27% of the control.

Similar to the root and shoot length, the best results of seedling length were related to the Zn NPs treated samples. After 48 h (Tables 6 and 8), Ag and Fe had seedling lengths less than the control. For the $AgNO_3$ primed samples, a 50 ppm concentration resulted in maximum seedling length and 100 ppm resulted in no seedling. Moreover, $MgSO_4$ showed the best increasing effect in seedling length. After the first week (Tables 7 and 9), in the case of nano-primed seeds, most of the treatments had seedling lengths more than the control. Zinc acetate and $MgSO_4$ were most effective in increasing the seedling length and the minimum was related to the 100 ppm $AgNO_3$ treatments.

**Table 7.** Effect of biogenic NPs priming on the seed germination parameters of flax on the seventh day under different nanoparticle concentrations. Data are presented as mean values ± SD for three independent experiments. The same letters within a column showed no significant difference at a 95% probability level at the $p < 0.05$ level.

| NP | Concentration (ppm) | Germination (%) | Shoot Length (cm) | Root Length (cm) | Seedling Length (cm) | Root to Shoot Ratio | SVI | SLSI (%) | RLSI (%) |
|---|---|---|---|---|---|---|---|---|---|
| Non (Control) | 0 | 93 ± 2 [a] | 3.9 ± 0.2 [d] | 2.7 ± 0.7 [h] | 6.7 ± 0.7 [h] | 0.7 ± 0.2 [f] | 622 ± 70 [g] | 100 ± 0 [c] | 100 ± 0 [h] |
| C-ZnO | 50 | 87 ± 2 [b] | 4.4 ± 0.6 [c] | 4.5 ± 0.6 [e] | 8.9 ± 1.1 [ef] | 1.0 ± 0.1 [c] | 769 ± 96 [e] | 112 ± 16 [bc] | 164 ± 21 [ef] |
| | 100 | 87 ± 2 [b] | 4.4 ± 0.3 [c] | 5.5 ± 0.7 [d] | 9.8 ± 0.6 [de] | 1.2 ± 0.2 [b] | 849 ± 54 [d] | 115 ± 7 [bc] | 195 ± 28 [cd] |
| | 150 | 97 ± 2 [a] | 4.4 ± 0.4 [c] | 6.0 ± 0.1 [cd] | 10.3 ± 0.6 [d] | 1.3 ± 0.1 [b] | 999 ± 63 [b] | 114 ± 9 [bc] | 214 ± 15 [c] |
| NC-ZnO | 50 | 80 ± 3 [bc] | 5.4 ± 0.2 [a] | 7.5 ± 1.6 [b] | 12.9 ± 1.4 [b] | 1.4 ± 0.4 [b] | 1033 ± 110 [ab] | 137 ± 6 [b] | 273 ± 57 [b] |
| | 100 | 80 ± 2 [bc] | 4.6 ± 0.5 [bc] | 6.0 ± 0.4 [cd] | 10.6 ± 0.7 [d] | 1.3 ± 0.1 [b] | 850 ± 60 [d] | 118 ± 12 [bc] | 218 ± 15 [c] |
| | 150 | 83 ± 3 [bc] | 4.1 ± 0.2 [cd] | 4.6 ± 0.6 [e] | 8.7 ± 0.6 [f] | 1.1 ± 0.2 [bc] | 725 ± 52 [ef] | 105 ± 6 [c] | 166 ± 24 [ef] |
| Zn | 50 | 87 ± 2 [b] | 5.0 ± 0.2 [ab] | 8.2 ± 0.9 [a] | 13.2 ± 1.2 [a] | 1.6 ± 0.1 [a] | 1148 ± 109 [a] | 128 ± 10 [ab] | 300 ± 31 [a] |
| | 100 | 73 ± 3 [d] | 5.0 ± 0.0 [ab] | 6.4 ± 1.2 [c] | 11.4 ± 1.1 [c] | 1.3 ± 0.3 [b] | 834 ± 81 [d] | 128 ± 10 [ab] | 232 ± 45 [bc] |
| | 150 | 73 ± 3 [d] | 4.6 ± 1.2 [bc] | 5.6 ± 0.6 [d] | 10.2 ± 0.6 [d] | 1.3 ± 0.6 [b] | 752 ± 47 [e] | 118 ± 30 [bc] | 204 ± 23 [cd] |
| MgO | 50 | 76 ± 4 [d] | 4.8 ± 0.5 [b] | 4.6 ± 0.6 [e] | 9.4 ± 0.7 [e] | 1.0 ± 0.2 [c] | 719 ± 57 [ef] | 122 ± 13 [b] | 168 ± 23 [ef] |
| | 100 | 80 ± 3 [bc] | 4.4 ± 0.5 [bc] | 3.9 ± 1.1 [fg] | 8.4 ± 1.5 [f] | 0.9 ± 0.2 [e] | 662 ± 120 [fg] | 114 ± 12 [bc] | 141 ± 40 [fg] |
| | 150 | 73 ± 3 [d] | 5.0 ± 0.4 [ab] | 4.1 ± 0.7 [ef] | 9.1 ± 0.7 [e] | 0.8 ± 0.2 [e] | 669 ± 55 [f] | 128 ± 10 [b] | 150 ± 27 [f] |
| Ag | 50 | 90 ± 2 [ab] | 5.4 ± 0.6 [a] | 4.9 ± 0.2 [e] | 10.2 ± 0.5 [d] | 0.9 ± 0.1 [d] | 923 ± 45 [c] | 137 ± 16 [a] | 177 ± 9 [de] |
| | 100 | 83 ± 3 [bc] | 5.2 ± 0.6 [a] | 4.7 ± 1.3 [e] | 10.0 ± 1.5 [d] | 0.9 ± 0.3 [d] | 831 ± 127 [d] | 134 ± 16 [a] | 173 ± 48 [e] |
| | 150 | 81 ± 2 [bc] | 4.8 ± 1.2 [b] | 4.4 ± 0.7 [e] | 9.1 ± 1.8 [e] | 1.0 ± 0.2 [cd] | 739 ± 149 [e] | 122 ± 32 [bc] | 159 ± 27 [f] |
| Cu | 50 | 90 ± 2 [ab] | 4.6 ± 0.5 [bc] | 5.9 ± 0.6 [c] | 10.5 ± 0.9 [d] | 1.3 ± 0.2 [b] | 945 ± 82 [bc] | 118 ± 12 [bc] | 214 ± 23 [c] |
| | 100 | 76 ± 4 [d] | 3.9 ± 0.5 [d] | 2.6 ± 0.5 [h] | 6.5 ± 0.4 [h] | 0.7 ± 0.2 [f] | 498 ± 31 [i] | 99 ± 12 [cd] | 95 ± 17 [hi] |
| | 150 | 77 ± 4 [d] | 4 ± 0.3 [c] | 2.8 ± 0.8 [h] | 7.1 ± 0.6 [g] | 0.7 ± 0.2 [f] | 546 ± 48 [h] | 109 ± 7 [c] | 104 ± 31 [h] |
| Fe | 50 | 86 ± 2 [b] | 3.4 ± 0.5 [e] | 2.4 ± 0.6 [i] | 5.7 ± 1.0 [i] | 0.7 ± 0.1 [f] | 498 ± 90 [i] | 86 ± 12 [e] | 86 ± 23 [i] |
| | 100 | 76 ± 4 [d] | 4.1 ± 0.6 [cd] | 2.7 ± 0.6 [h] | 6.9 ± 0.7 [gh] | 0.7 ± 0.2 [f] | 537 ± 57 [h] | 106 ± 16 [c] | 100 ± 23 [h] |
| | 150 | 87 ± 2 [b] | 4.1 ± 0.5 [cd] | 2.4 ± 0.8 [i] | 6.5 ± 0.0 [h] | 0.6 ± 0.2 [g] | 553 ± 0 [h] | 105 ± 12 [c] | 86 ± 17 [i] |
| *p*-value | | 0.0000 | 0.0100 | 0.0007 | 0.0001 | 0.0033 | 0.0082 | 0.0004 | 0.0201 |

The same letters within a column showed no significant difference at a 95% probability level at the $p < 0.05$ level. ppm: part per million, nm: nanometer, cm: centimeter, SD: standard deviation, *p*-value: probability value, SVI: seedling vigor index, SLSI: shoot length stress tolerance index, RLSI: root length stress tolerance index.

**Table 8.** Effect of metal salt (precursors) priming on seed germination parameters of flax on the second day under different nanoparticle concentrations. Data are presented as mean values ± SD for three independent experiments. The same letters within a column showed no significant difference at a 95% probability level at the $p < 0.05$ level.

| Metal Salt | Concentration (ppm) | Germination (%) | Shoot Length (cm) | Root Length (cm) | Seedling Length (cm) | Root to Shoot Ratio | SVI | SLSI (%) | RLSI (%) |
|---|---|---|---|---|---|---|---|---|---|
| Non (Control) | 0 | 87 ± 2 [bc] | 0.3 ± 0.0 [d] | 2.2 ± 0.3 [b] | 2.6 ± 0.2 [cd] | 7.1 ± 1.3 [a] | 221 ± 24 [bc] | 100 ± 0 [c] | 100 ± 0 [bc] |
| Zn (CH$_3$ CO$_2$)$_2$ | 50 | 97 ± 2 [a] | 0.5 ± 0.1 [a] | 2.6 ± 0.3 [ab] | 3.1 ± 0.3 [b] | 5.3 ± 1.5 [c] | 303 ± 31 [a] | 164 ± 34 [a] | 117 ± 16 [b] |
| | 100 | 93 ± 2 [ab] | 0.5 ± 0.1 [b] | 2.6 ± 0.3 [ab] | 3.0 ± 0.1 [b] | 5.4 ± 0.8 [c] | 283 ± 30 [ab] | 147 ± 26 [b] | 114 ± 13 [b] |
| | 150 | 91 ± 2 [ab] | 0.4 ± 0.1 [bc] | 2.1 ± 0.9 [b] | 2.5 ± 0.7 [cd] | 5.3 ± 2.1 [c] | 228 ± 79 [bc] | 131 ± 26 [bc] | 94 ± 39 [c] |
| MgSO$_4$ | 50 | 83 ± 3 [bc] | 0.5 ± 0.0 [a] | 2.5 ± 0.2 [ab] | 3.0 ± 0.2 [b] | 4.9 ± 0.8 [d] | 255 ± 19 [b] | 164 ± 14 [a] | 112 ± 11 [b] |
| | 100 | 83 ± 2 [bc] | 0.4 ± 0.1 [bc] | 2.8 ± 0.6 [a] | 3.2 ± 0.9 [ab] | 7.2 ± 1.5 [a] | 266 ± 53 [b] | 125 ± 38 [bc] | 125 ± 25 [a] |
| | 150 | 87 ± 2 [bc] | 0.5 ± 0.2 [a] | 2.7 ± 0.3 [ab] | 3.3 ± 0.3 [a] | 5.6 ± 2.3 [bc] | 282 ± 30 [ab] | 169 ± 53 [a] | 121 ± 14 [ab] |
| AgNO$_3$ | 50 | 87 ± 2 [bc] | 0.5 ± 0.1 [a] | 2.8 ± 0.6 [a] | 3.4 ± 0.5 [a] | 6.0 ± 3.0 [b] | 297 ± 46 [a] | 168 ± 48 [a] | 128 ± 27 [a] |
| | 100 | 67 ± 2 [d] | 0.3 ± 0.3 [e] | 0.6 ± 0.3 [ef] | 0.8 ± 0.6 [fg] | 2.0 ± 0.9 [f] | 54 ± 41 [f] | 81 ± 98 [d] | 25 ± 14 [fg] |
| | 150 | 0 [e] | 0 [f] | 0 [g] | 0 [h] | 0 [g] | 0 [g] | 0 [f] | 0 [h] |

**Table 8.** *Cont.*

| Metal Salt | Concentration (ppm) | Germination (%) | Shoot Length (cm) | Root Length (cm) | Seedling Length (cm) | Root to Shoot Ratio | SVI | SLSI (%) | RLSI (%) |
|---|---|---|---|---|---|---|---|---|---|
| CuSO$_4$ | 50 | 87 ± 2 [bc] | 0.4 ± 0.1 [bc] | 1.3 ± 0.3 [d] | 1.7 ± 0.3 [e] | 3.3 ± 0.9 [e] | 147 ± 25 [d] | 125 ± 23 [bc] | 58 ± 13 [e] |
| | 100 | 94 ± 3 [ab] | 0.2 ± 0.2 [e] | 0.7 ± 0.1 [e] | 0.9 ± 0.2 [f] | 3.3 ± 1.9 [e] | 86 ± 24 [e] | 69 ± 51 [e] | 31 ± 5 [f] |
| | 150 | 90 ± 3 [b] | 0.3 ± 0.1 [d] | 0.6 ± 0.2 [ef] | 0.9 ± 0.1 [f] | 2.0 ± 0.8 [f] | 82 ± 14 [e] | 100 ± 26 [c] | 26 ± 7 [fg] |
| FeCl$_3$ | 50 | 90 ± 2 [b] | 0.5 ± 0.1 [b] | 2.4 ± 0.3 [b] | 2.8 ± 0.3 [c] | 5.5 ± 2.2 [c] | 255 ± 26 [b] | 144 ± 28 [b] | 105 ± 16 [bc] |
| | 100 | 91 ± 3 [ab] | 0.4 ± 0.1 [c] | 1.8 ± 0.2 [c] | 2.2 ± 0.5 [d] | 5.0 ± 1.4 [cd] | 198 ± 14 [c] | 120 ± 6 [bc] | 81 ± 9 [d] |
| | 150 | 87 ± 2 [bc] | 0.5 ± 0.1 [b] | 2.6 ± 0.3 [ab] | 3.0 ± 0.4 [b] | 5.9 ± 1.1 [b] | 264 ± 36 [b] | 144 ± 42 [b] | 115 ± 13 [b] |
| *p*-value | | 0.0111 | 0.0001 | 0.0008 | 0.0039 | 0.0007 | 0.0004 | 0.0100 | 0.0006 |

The same letters within a column showed no significant difference at a 95% probability level at the $p < 0.05$ level. ppm: part per million, nm: nanometer, cm: centimeter, SD: standard deviation, *p*-value: probability value, SVI: seedling vigor index, SLSI: shoot length stress tolerance index, RLSI: root length stress tolerance index.

**Table 9.** Effect of metal salt (precursors) priming on seed germination parameters of flax on the seventh day under different nanoparticle concentrations. Data are presented as mean values ± SD for three independent experiments. The same letters within a column showed no significant difference at a 95% probability level at the $p < 0.05$ level.

| Metal Salt | Concentration (ppm) | Germination (%) | Shoot Length (cm) | Root Length (cm) | Seedling Length (cm) | Root to Shoot Ratio | SVI | SLSI (%) | RLSI (%) |
|---|---|---|---|---|---|---|---|---|---|
| Non (Control) | 0 | 93 ± 2 [a] | 3.9 ± 0.2 [de] | 2.7 ± 0.7 [h] | 6.7 ± 0.7 [g] | 0.7 ± 0.16 [e] | 622 ± 70 [f] | 100 ± 0 [d] | 100 ± 0 [f] |
| Zn (CH$_3$ CO$_2$)$_2$ | 50 | 87 ± 2 [b] | 5.9 ± 0.7 [a-c] | 7.1 ± 1.4 [a] | 13.0 ± 1.8 [a] | 1.2 ± 0.2 [bc] | 112 ± 154 [i] | 151 ± 12 [b] | 259 ± 52 [a] |
| | 100 | 87 ± 2 [b] | 4.2 ± 0.5 [d] | 5.5 ± 0.70 [d] | 9.7 ± 0.6 [d] | 1.3 ± 0.3 [b] | 844 ± 55 [de] | 109 ± 12 [d] | 200 ± 18 [c] |
| | 150 | 80 ± 2 [bc] | 4.2 ± 0.9 [d] | 5.9 ± 1.1 [c] | 10.1 ± 1.4 [cd] | 1.4 ± 0.4 [b] | 810 ± 114 [de] | 109 ± 22 [d] | 214 ± 40 [bc] |
| MgSO$_4$ | 50 | 83 ± 3 [bc] | 5.0 ± 0.3 [c] | 5.8 ± 1.2 [c] | 10.8 ± 1.1 [c] | 1.2 ± 0.3 [c] | 901 ± 88 [c] | 128 ± 10 [c] | 211 ± 44 [bc] |
| | 100 | 83 ± 3 [bc] | 6.7 ± 3.3 [a] | 6.4 ± 2.5 [b] | 13.1 ± 3.8 [a] | 1.1 ± 0.6 [c] | 1093 ± 314 [a] | 173 ± 84 [a] | 232 ± 89 [b] |
| | 150 | 90 ± 2 [a] | 5.1 ± 1.2 [c] | 5.7 ± 0.5 [c] | 10.9 ± 0.9 [c] | 1.2 ± 0.4 [bc] | 978 ± 85 [bc] | 131 ± 32 [bc] | 209 ± 18 [bc] |
| AgNO$_3$ | 50 | 87 ± 2 [b] | 5.6 ± 0.5 [bc] | 6.2 ± 0.6 [b] | 11.9 ± 0.6 [b] | 1.1 ± 0.2 [c] | 1029 ± 54 [b] | 144 ± 12 [b] | 227 ± 23 [b] |
| | 100 | 40 ± 5 [e] | 1.0 ± 0.7 [g] | 1.9 ± 1.9 [i] | 2.9 ± 2.6 [i] | 1.7 ± 1.0 [a] | 116 ± 103 [i] | 25 ± 19 [g] | 70 ± 68 [g] |
| | 150 | 0 [f] | 0 [h] | 0 [k] | 0 [j] | 0 [h] | 0 [j] | 0 [h] | 0 [i] |
| CuSO$_4$ | 50 | 77 ± 4 [c] | 5.0 ± 0.4 [c] | 3.5 ± 0.7 [g] | 8.5 ± 0.7 [e] | 0.7 ± 0.2 [e] | 651 ± 54 [f] | 128 ± 10 [c] | 127 ± 26 [ef] |
| | 100 | 74 ± 3 [cd] | 3.4 ± 0.8 [e] | 4.6 ± 0.3 [e] | 4.0 ± 0.6 [h] | 0.2 ± 0.1 [g] | 302 ± 46 [h] | 86 ± 22 [e] | 27 ± 10 [h] |
| | 150 | 80 ± 3 [bc] | 3.1 ± 0.2 [ef] | 0.7 ± 0.3 [j] | 4.0 ± 0.5 [h] | 0.2 ± 0.1 [g] | 310 ± 38 [h] | 80 ± 6 [ef] | 27 ± 10 [h] |
| FeCl$_3$ | 50 | 81 ± 2 [bc] | 5.7 ± 0.5 [bc] | 5.0 ± 0.7 [d] | 10.7 ± 1.2 [c] | 0.9 ± 0.0 [d] | 870 ± 96 [d] | 147 ± 13 [b] | 182 ± 26 [d] |
| | 100 | 71 ± 1 [cd] | 5.2 ± 0.5 [c] | 2.9 ± 1.1 [h] | 8.1 ± 0.5 [ef] | 0.6 ± 0.3 [ef] | 574 ± 33 [g] | 134 ± 22 [bc] | 104 ± 40 [f] |
| | 150 | 80 ± 3 [bc] | 6.0 ± 0.6 [ab] | 4.2 ± 0.9 [ef] | 10.2 ± 0.9 [cd] | 0.7 ± 0.2 [e] | 823 ± 69 [de] | 154 ± 15 [b] | 154 ± 31 [de] |
| *p*-value | | 0.0044 | 0.0400 | 0.0006 | 0.0007 | 0.0055 | 0.0050 | 0.0001 | 0.0001 |

The same letters within a column showed no significant difference at a 95% probability level at the $p < 0.05$ level. ppm: part per million, nm: nanometer, cm: centimeter, SD: standard deviation, *p*-value: probability value, SVI: seedling vigor index, SLSI: shoot length stress tolerance index, RLSI: root length stress tolerance index.

In the early stages of flax seedling, the shoot grew with a higher speed in comparison with the root. Considering the results of Tables 6 and 8, on the second day, all of the R/S values were less than the control in NP and the salt primed samples, and on the seventh day, most of the samples had a R/S more than the control.

## 4. Discussion

### 4.1. Effect of Biogenic NPs and Their Counterpart Salts on Physical Characteristics of Wheat Seedling

As seed germination is the first step to start a successful crop improvement, it could be considered as an index to assay the enhancive or inhibitive effect of newly developed agrochemicals such as nanomaterials [19].

In present study, the seeds were exposed to the NPs or salts only for 12 h, but the priming effect was observed up to several days. In this regard, it could be suggested that the NPs or metal ions are absorbed on the surface of the seeds and gradually release to show their effect during a period of seven days. Additionally, the reason of difference between the response of nano-primed seeds with salt-primed ones may be illustrated as the result of a gradual release of ions from NPs by sub-toxic levels rather than the exposure to a large number of ions in the case of priming with metal salts, which may cause stress in the germination process [5].

Previous findings have reported that zinc nanoparticulate priming was more effective than zinc salts in enhancing the seedling growth. For instance, it was found that Zn NP treated wheat seeds surpassed elemental Zn values over $ZnSO_4$, indicating that NPs are more efficient at delivering Zn to plant tissues than $ZnSO_4$, which suggests it is carried out during a particle-specific mechanism [26]. Similar studies have shown that accumulation of Zn from NP treatment was more than the predicted values upon dissolved Zn concentration [19]. However, Zn is an essential metal for plant growth, but may be a phytotoxic metal when it exceeds the tolerance limit, depending on the plant species or plant's studied part [27].

Previous studies have also reported similar results to our findings on the seedling growth of ZnO treated wheat seeds [20,28]. In the study conducted by Ahmed et al., C-ZnO NPs with very high concentrations of 0.05, 0.5, 2, and 5 mg/mL were applied on four different seeds such as radish, cucumber, tomato, and alfalfa to study the toxicity effect of NPs on seeds. They reported that C-ZnO exhibited no obvious toxic effect on the germination, root, and shoot growth of these seeds [19]. Similarly, in our study, C-ZnO and NC-ZnO NP treatment improved the seedling growth. Several mechanisms can be found in the literature illustrating the various effects of NPs on plant parts and cell reactions. For example, the effect of NPs on specific enzymatic reactions and different enzymes such as amylase could elucidate the effect of NPs on seed germination. It is not clear at this point whether NP toxicity is stimulated by the particles or dissolved ions [8]. The effect of NPs may also be due to the interaction of NPs with some parts of the plants such as the cell wall or membrane components. The size of the NPs is consistent with the structure of the plant cell wall to enter the cell at the point that the accumulation of reactive oxygen species (ROS) can be started [27]. ROS can influence the permeability of the cells as it interferes with the plasma membrane. Consequently, more NPs can result in intense stress after reaching the cells and stimulating the formation of stress-induced secondary metabolites [27].

In another study, Ag NP toxicity on rockcress seeds was shown to be dependent on the size and concentration [5]. Ag NPs with a size of 80 nm were only deteriorative at higher concentrations and those of 20 and 40 nm resulted in severe root growth inhibition. The researchers supposed that Ag NPs apoplastically transported through the root tissues [5]. The inhibitory effect of Ag NPs on the germination index was also seen in the case of cucumber. Similar results were also reported by Vanninia et al. [15] as a 10 ppm concentration of Ag NPs adversely influenced the seedling growth of wheat seeds. They also reported an induction of morphological modifications in the root tip cells by Ag NPs. According to the microscopy of the treated seed roots, Ag NPs did not enter the root cells and were located in the outer cells of the root cup. It was suggested by TEM analysis that the toxicity effect of Ag NPs resulted from the release of $Ag^+$ ions from Ag NPs [15]. Abbasi Khalaki et al. [29] reported an enhancement in seedling growth of *thymus kotschyanus* seeds treated with 20 and 60% concentrations of Ag NPs [29].

The results reported by Zakharova et al. [30] were comparable with our obtained results, in which wheat seeds were soaked in the presence of CuO NPs. They also reported that exposure of wheat seeds to 10 ppm CuO NPs showed a 14.5% improvement in germination and a two-fold increase in root and shoot length in comparison with the control. At higher concentrations of CuO NPs, both stimulation and toxic effects were observed (decline in root length) [30].

The effect of wheat seed treatment with Cu NPs on germination and seedling vigor index was studied by Yasmeen et al. under laboratory conditions [31]. Germination percentage, root, and shoot lengths were calculated and the results indicated that exposure of wheat seeds to Cu NPs led to a decline in germination percentage and severe reduction in root and shoot length. Therefore, Cu NPs adversely affected the germination and growth of wheat seeds [31]. This substantial decrease in the plantlet growth is consistent with previous wheat field studies, where the application of excessive NPs resulted in reduced plantlet length and distorted plantlet physiology [32].

Plaksenkova et al. [33] studied the effect of $Fe_3O_4$ NPs stress on the growth and development of rocket seeds. According to the results, 1 ppm, 2 ppm, and 4 ppm concentrations of $Fe_3O_4$ NPs have a positive effect on the growth and development of rocket seedlings. In a similar study, Yi Hao et al. [34] studied the effects of nano-priming with different $Fe_2O_3$ morphologies such as $Fe_2O_3$ nanocubes, $Fe_2O_3$ short nanorods, and $Fe_2O_3$ long nanorods at the concentrations of five to 150 ppm on rice seeds during the germination. They found that all NPs considerably stimulated the root growth, and promoted shoot length at most concentrations while $Fe_2O_3$ long nanorods inhibited the seed germination significantly and showed a different biological effect from other $Fe_2O_3$ nanomaterials due to their different shape [34].

Ngo et al. [35] studied the effects of Fe NPs on soybean germination, growth, crop yield, and product quality. In laboratory conditions, the germination rates of soybean seeds soaked with Fe NPs was 80% at the concentration of 0.08 g/ha, while germination of the control sample was 55% [35]. Consequently, various data obtained from such experiments could really be helpful in developing priming treatments for field experiments and other agricultural purposes.

### 4.2. Effect of Biogenic NPs and Their Counterpart Salts on Physical Characteristics of Flax Seedling

The special structure of the flax seed's coat was the reason for we chose this plant for our experiments. The envelope or testa of the flax seed contains about 15% of mucilage, which mainly contains distinct types of arabinoxylans and water-soluble hydrocolloid/polysaccharides that contribute to its gel qualities by forming large aggregates in solution [36]. It is suggested that nano-priming of the flax seeds as well as the NP behavior may be affected by the mucilage of the seed coat due to the thick chemical environment in which NPs are trapped. Therefore, according to the obtained results, it could be considered as one the reasons that we observed different results of flax seed seedlings in comparison with wheat seeds.

There are a few works that have studied the influence of NPs on the seedling parameters of flax seeds. In support of our results, the effect of different metal and metal oxide NPs have been presented. As an example, it was reported that the application of biosynthesized MgO NPs enhanced the seed germination and growth parameters of peanut seeds compared with the control. The authors, using physicochemical methods including UV and SEM analyses, indicated that the MgO NPs penetrated into the seed coat, support water uptake inside the seeds, and then affecting the seed germination and growth rate mechanism [37].

In contrast to the results obtained from our research, Gorczyca et al. [17] noticed that 100 ppm of Ag NP treatments applied to flax seeds had a limited effect on the germination and early development of the seedlings in comparison with the control. The response of the flax seeds to the NPs was reported as an increase in chlorophyll content [17]. Zaeem et al. [27] investigated the effect of green synthesized C-ZnO NPs at concentrations of 0, 1, 10, 100, 500, and 1000 ppm on the growth of flax seeds. All of the treated flax seeds had a root development of different lengths ranging from 2.62 cm (for 1000 ppm ZnO NPs) to 7.08 cm (for 10 ppm ZnO NPs) with 3.85 cm for the control. These results indicate the efficiency of different concentrations of ZnO NPs in seed germination. At a ZnO NP concentration of above 10 ppm, the higher the concentration of NPs, the lower the root

length. The increased sensitivity of radicles to NPs is due to the large surface area of the NPs. They suggested that the observed inhibitory effect on seed germination may be due to the very small size of NPs and the dissolution power of ZnO to $Zn^{2+}$ ions [27].

To the best of our knowledge, there has not been any report on the positive or negative effect of Cu NPs on flax seedling for a comparison with our findings. In previously reported research, despite our findings, stimulating the effects of Fe NPs on the seedlings of different species has been described, for example, in rocket [8], rice [34], and soybean [35]. Clearly, different results were obtained from flax seed priming in comparison to the wheat seeds in the germination and seedling growth parameters.

*4.3. A Comparison between the Effect of Biogenic NPs and Their Counterpart Salts on Physiological Characteristics of Wheat and Flax Seedlings*

On the seventh day of the experiment, in a comparison between the effect of studied NPs in applied concentrations, the most effective one in shoot and root development was Zn NPs and the less effective NPs were Ag for wheat and Fe for flax seeds (Figure 2). This shows that Zn NPs were not toxic at the applied concentrations and even showed aa stimulating effect on both wheat and flax seeds. Vice versa, the applied concentrations of Ag NPs were toxic for wheat, but stimulating for flax. As all of the experiment conditions were the same for all of the samples, it could be concluded that these differences are related to the seed species. Among the tested metal slats, zinc acetate had the most stimulating effect and $CuSO_4$ was the most toxic for both flax and wheat. Nanoforms of metals and metal oxides have been reported to significantly improve root or shoot elongation and the seed germination of wheat in comparison with bulk materials [38]. This kind of growth development mainly depends on the concentration of NP, duration of nano-priming, growth medium, and species of plant [19].

Among the measured parameters, root length is more sensitive than shoot length. Between wheat and flax roots, the flax root length was more sensitive against NP and salt treatments and wheat shoot length was the less sensitive parameter. Overall, flax seeds were more sensitive to the treatments compared to the wheat seeds. Although the factors that impact the root and shoot elongation following NP exposure are not clear yet, it could be suggested that the polymeric network of flax seed mucilage traps NPs or metal ions, and that their accessibility for flax seeds differed from the wheat seeds in the period of our experiment [19].

In the majority of cases, wheat seeds had more G% than flax seeds. A total of 150 ppm of $AgNO_3$ solution had such a toxic effect on flax seeds that no germination was observed in this treatment. Similarly, metal-based NPs have been reported to show dual impacts on plant growth such as seed germination. Positive effects of metal-based NP treatments were displayed in different plants [5]. Seed germination of soybean seeds was enhanced by nano-priming with Co, Fe, and Cu NPs [36]; similar findings were also reported in the case of some Solanaceae crops after treatment with ZnO and $TiO_2$ NPs [39]. The obtained results were comparable with those reported by Feizi et al. [38], in which seed treatment with $TiO_2$ NPs at low concentrations (1–2 ppm) resulted in an improvement in the germination of wheat seeds as well as seedling elongation compared to untreated wheat, but no significant effect at a concentration of 100 ppm was observed [38,40].

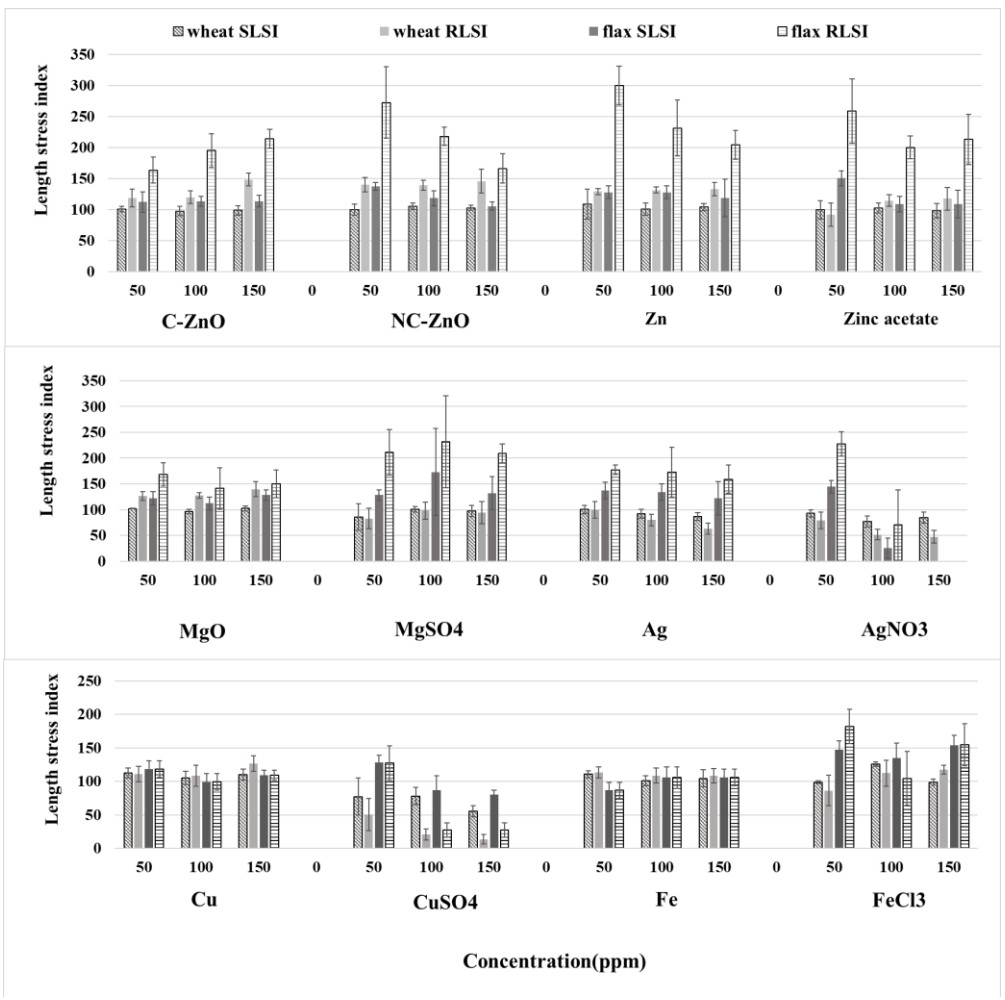

**Figure 2.** Dose response effect of nanoparticles and their correspondent metal salts on the shoot and root stress tolerance index (SLSI and RLSI, respectively) of wheat and flax seeds on the seventh day of the experiment and their corresponding standard deviation. ppm: part per million.

## 5. Conclusions

With intensive use of traditional farming, adequate yield can be attained in large scale cropping systems, but natural resources are simultaneously exhausted, biodiversity is diminished, and ecosystem balance deteriorates due to air, water and soil pollution, leading to irreversible problems. It is suggested that using green synthesized nano-size minerals for seed treatment could be helpful in seedling growth improvement. The present study provides new information on the possible positive or toxic effects of seed priming with NPs on the germination percentages, shoot length, root length, seedling length, root/shoot ratio, seedling vigor index (SVI), shoot length stress tolerance index (SLSI), and root length stress tolerance index (RLSI), which were calculated on the second and seventh days on two popular early growth plants: wheat and flax. The plant's dual responses varied among NP type and correlated to the tested concentrations. According to the obtained results, the response of the tested plants to a certain NP was different between flax and wheat. Moreover, it differed between the applied concentrations of the NPs. For example, Ag NPs showed a significant positive effect on the root and shoot elongation of flax seedlings, but there was a dose dependent decrease in the root and shoot elongation of wheat seedlings over their respective controls. Another important result is that flax seed was more sensitive to priming with metal salts and NPs in comparison with wheat seed. Furthermore, the influence of these treatments was investigated in earlier stages of the growth, on the second day of the experiment, in comparison with the seventh day of the experiment. Among

the studied NPs, Zn and Ag NPs exhibited the best biological effects on the growth and development of wheat and flax, respectively. The effect of the nanoparticle's counterpart metal salts on the seedling parameters was also studied for a comparison with those of the nanoparticulate. Overall, nanoparticle treatments were more effective than the metal salt treatments in root and shoot development.

The basic mechanisms of the effect of NPs on seed germination and growth development or inhibition need to be investigated in future investigations.

The above findings could be further used for agricultural applications to increase seed germination and promote the overall yield. Increased seed germination and early plant growth is really vital in achieving crop productivity. The extensive effect on the plant growth's early stages might be followed by similar effects at later stages, so by applying nanoparticles in this way, we may be able to improve plant productivity. Furthermore, it can be concluded that biogenic NPs can be applied as novel nanoparticles to improve the growth of wheat and flax as well as to decrease the application of conventional agricultural fertilizers, helping the promotion of sustainable agriculture. The positive or negative effects of biogenic NPs have been reported on different plants and still need to be experienced in large scales such as field or greenhouse experiments. The results also suggest that the release of different NPs into the environment may have negative effects on plant communities.

**Author Contributions:** Manuscript conception, M.B. and M.Z.; Methodology, M.B.; Data analysis, M.B. and M.Z.; Validation and investigation, S.I.S.; Writing—original draft preparation, M.Z. and M.B.; Writing—review and editing, K.M.-S.M.; Project administration, M.R.N. All authors have read and agreed to the published version of the manuscript.

**Funding:** This research received no external funding.

**Data Availability Statement:** The datasets used and/or analyzed during the current study are available from the corresponding author on reasonable request.

**Acknowledgments:** This paper was supported by the Kadyrov Chechen State University development program 2021–2030.

**Conflicts of Interest:** The authors declare no conflict of interest.

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
