# Peer review of "Ameliorating Seed Germination and Seedling Growth of Nano-Primed Wheat and Flax Seeds Using Seven Biogenic Metal-Based Nanoparticles"

_agronomy, doi:10.3390/agronomy12040811_

Round 1

Reviewer 1 Report

This research manuscript discusses the effect of different concentrations of several priming substances (biogenic nanoparticles – NPs, and their counterpart metallic salts) on germination and seedling traits of wheat and flax grown in vitro.

The results showed a different response of wheat and flax to NPs applications and also in relation to their concentrations; NPs treatments were more effective than metal salt ones in term of root and shoot growth.

In general, the manuscript meets the scopes of the Journal and gives a lot of informations useful for practical application.

However, major changes are needed as reported below.

  • Results. Lines from 186 to 195 can be eliminated because the same informations are present in “Materials and Methods” section.
  •  The tables are unclear and difficult to read. It is necessary to reduce their number and, moreover, the decimal places of the reported values can be reduced. In the figures SD values are missing.
  • The statistical analysis in the Tables 2 to 9 should be re-done adopting two-way ANOVA; for each crop and for each date of observation (2nd and 7th days of the experiment) it is useful to report the mean effects (significance) of the priming substances (NPs and counterpart metallic salts, all together) as well as the mean effects of applied concentrations and finally the interaction between the two factors. Thereby the ‘Results’ section should be revised.
  • Citations and References. The citations and the ‘References’ must be revised according Agronomy Standard (i.e. numerated and reported in chronological order instead of alphabetical one).

Author Response

Dear Reviewer

We gratefully acknowledge the detailed revision of the text and useful suggestions to improve the paper by the reviewers. We have closely followed he/she suggestions and introduced the required changes in the text. Main changes are highlighted into the manuscript in YELLOW. Below, we have included reviewer comments and our responses.

  • According to the reviewer suggestion some sentences omitted and some parts summarized.
  • All of the tables simplified
  • In the figures SD values are added.
  • The applied method was two-way ANOVA but it was written one-way ANOVA (in materials and methods part) by mistake. Already revised.
  • Mean comparisons were conducted for each crop and for each date of observation. Result section revised.
  • Citations and References revised.

We hope that after these enhancements the manuscript can now be accepted, although we are certainly willing to consider further changes if necessary.

Yours sincerely

Reviewer 2 Report

This article,” Ameliorating Seed Germination and Seedling of Nano-Primed 2 Wheat and Flax Seeds by the Using Seven Biogenic Metal- 3 Based Nanoparticles” deals with the effect of seven different nanoparticles on the germination of two plants, Wheat and Flax. Nanobiotechnology is one of the emerging techniques that can be helpful to achieve SDG about hunger and poverty. Biogenic-based nanoparticles are less toxic, cheap and more effective in alleviating the adverse environmental impacts on economically important crops. The significance of germination is understandable, but it is  based on only germination data. I would suggest authors to do one more pot experiment and use the best concentration of the best nanoparticle (based on the Petri plate experiments) to evaluate morpho-physio and biochemical responses. This will give a clear insight into the mechanism of action of these NPs on plant growth. Too many figures and tables of preliminary data are presented in the current MS. These preliminary data can be presented in supplementary files if can do more experiments.

Author Response

Dear Reviewer

We gratefully acknowledge the detailed revision of the text and useful suggestions to improve the paper by the reviewers. We have closely followed he/she suggestions and introduced the required changes in the text. Main changes are highlighted into the manuscript in YELLOW. Below, we have included reviewer comments and our responses.

It is a good idea to expand the results of our preliminary experiments to larger scales such as pot, greenhouse or field experiment. Moreover, we tried foliar spraying of Ag nanoparticles in a field scale and the results are more than that to be added to this article and it is better to be published as an individual paper.

We hope that after these enhancements the manuscript can now be accepted, although we are certainly willing to consider further changes if necessary.

Yours sincerely

Round 2

Reviewer 1 Report

After having carefully read the revision reports and the revised form of the manuscript, I’ve noticed that the Authors have made successfully efforts to improve their manuscript, taking into due consideration my remarks.

However, further minor corrections are required: 

since the two-way ANOVA procedure was applied, in the tables 2-9 authors should enter also the significance level of the two main effects, i.e. 1) priming substances (NPs and counterpart metallic salts) and 2) nanoparticle concentrations).

Author Response

Dear Editor

We gratefully acknowledge the detailed revision of the text and useful suggestions to improve the paper by the reviewers. We have closely followed he/she suggestions and introduced the required changes in the text. Main changes are highlighted into the manuscript in YELLOW. Below, we have included reviewer comments and our responses.

The applied method was two-way ANOVA but it was written one-way ANOVA (in materials and methods part) by mistake. Already revised.

Mean comparisons were conducted for each crop and for each date of observation. Result section revised.

We hope that after these enhancements the manuscript can now be accepted, although we are certainly willing to consider further changes if necessary.

Yours sincerely

Reviewer 2 Report

I understand that it's difficult to manage another experiment for the authors. However, data is not sufficient to publish in its current form. 

Author Response

(The authors gave the same response as above.)
